# Geological and taphonomic context for the new hominin species *Homo naledi* from the Dinaledi Chamber, South Africa

Paul HGM Dirks[1,2]*, Lee R Berger[2]*, Eric M Roberts[1,2], Jan D Kramers[3], John Hawks[2,4], Patrick S Randolph-Quinney[2,5], Marina Elliott[2,6], Charles M Musiba[2,7], Steven E Churchill[2,8], Darryl J de Ruiter[2,9], Peter Schmid[2,10], Lucinda R Backwell[2], Georgy A Belyanin[3], Pedro Boshoff[2,4], K Lindsay Hunter[2], Elen M Feuerriegel[2], Alia Gurtov[2,4], James du G Harrison[11], Rick Hunter[2], Ashley Kruger[2], Hannah Morris[2], Tebogo V Makhubela[3], Becca Peixotto[2,12], Steven Tucker[2]

[1]Department of Earth and Oceans, James Cook University, Townsville, Australia; [2]Evolutionary Studies Institute, National Centre for Excellence in PalaeoSciences, University of the Witwatersrand, Johannesburg, South Africa; [3]Department of Geology, University of Johannesburg, Johannesburg, South Africa; [4]Department of Anthropology, University of Wisconsin-Madison, Madison, United States; [5]School of Anatomical Sciences, University of the Witwatersrand, Johannesburg, South Africa; [6]Department of Archaeology, Simon Fraser University, Burnaby, Canada; [7]Department of Anthropology, University of Colorado Denver, Denver, United States; [8]Department of Evolutionary Anthropology, Duke University, Durham, United States; [9]Department of Anthropology, Texas A&M University, College Station, United States; [10]Anthropological Institute and Museum, University of Zurich, Zurich, Switzerland; [11]School of Animal, Plant and Environmental Sciences, University of the Witwatersrand, Johannesburg, South Africa; [12]Department of Anthropology, American University, Washington DC, United States

*For correspondence: paul.dirks@jcu.edu.au (PHGMD); Lee.Berger@wits.ac.za (LRB)

Competing interests: The authors declare that no competing interests exist.

Reviewing editors: Nicholas J Conard, University of Tübingen, Germany; Johannes Krause, University of Tübingen, Germany

**Abstract** We describe the physical context of the Dinaledi Chamber within the Rising Star cave, South Africa, which contains the fossils of *Homo naledi*. Approximately 1550 specimens of hominin remains have been recovered from at least 15 individuals, representing a small portion of the total fossil content. Macro-vertebrate fossils are exclusively *H. naledi*, and occur within clay-rich sediments derived from in situ weathering, and exogenous clay and silt, which entered the chamber through fractures that prevented passage of coarser-grained material. The chamber was always in the dark zone, and not accessible to non-hominins. Bone taphonomy indicates that hominin individuals reached the chamber complete, with disarticulation occurring during/after deposition. Hominins accumulated over time as older laminated mudstone units and sediment along the cave floor were eroded. Preliminary evidence is consistent with deliberate body disposal in a single location, by a hominin species other than *Homo sapiens*, at an as-yet unknown date.

**eLife digest** Modern humans, or *Homo sapiens*, are now the only living species in their genus. But as recently as 20,000 years ago there were other species that belonged to the genus *Homo*. Together with modern humans, these extinct human species, our immediate ancestors and their close relatives are collectively referred to as 'hominins'.

Now, Dirks et al. describe an unusual collection of hominin fossils that were found within the Dinaledi Chamber in the Rising Star cave system in South Africa. The fossils all belong to a newly discovered hominin species called *Homo naledi*, which is described in a related study by Berger et al. The unearthed fossils are the largest collection of hominin fossils from a single species ever to be discovered in Africa, and include the remains of at least 15 individuals and multiple examples of most of the bones in the skeleton.

Dirks et al. explain that the assemblage from the Dinaledi Chamber is unusual because of the large number of fossils discovered so close together in a single chamber deep within the cave system. It is also unusual that no other large animal remains were found in the chamber, and that the bodies had not been damaged by scavengers or predators. The fossils were excavated from soft clay-rich sediments that had accumulated in the chamber over time; it also appears that the bodies were intact when they arrived in the chamber, and then started to decompose.

Dirks et al. discuss a number of explanations as to how the remains came to rest in the Dinaledi Chamber, which range from whether *Homo naledi* lived in the caves to whether they were brought in by predators. Most of the evidence obtained so far is largely consistent with these bodies being deliberately disposed of in this single location by the same extinct hominin species. However, a number of other explanations cannot be completely ruled out and further investigation is now needed to uncover the series of events that resulted in this unique collection of hominin fossils.

## Introduction

The Pliocene-Pleistocene cave deposits in the Cradle of Humankind World Heritage Site (South Africa) preserve a diversity of hominin fossils in a varied set of contexts (*Hughes and Tobias, 1977*; *Clarke, 1998*; *Partridge et al., 2003*; *Berger et al., 2010*; *Bruxelles et al., 2014*; *Berger et al., 2015*). Hominin remains in the area are generally encased in lithified clastic deposits in caves that are situated in stromatolite-rich dolomite of the Malmani Subgroup (*Eriksson et al., 2006*) (*Figure 1*). Sedimentological and taphonomic descriptions of notable fossil sites (*Brain, 1981*; *de Ruiter et al., 2009*; *Dirks et al., 2010*; *Pickering and Kramers, 2010*; *Pickering et al., 2011a*, *2011b*) indicate that fossils were trapped and preserved in caves as a result of a range of processes including death traps, scavenging, mud flows and predation. Distribution patterns of fossiliferous caves in the area suggest that fossil deposition occurred in caves that are close to critical resources such as water (*Reynolds et al., 2011*; *Dirks and Berger, 2013*).

The cave deposits represent sediment traps that have collected fossils for at least 3 Ma, and possibly longer, thus, providing a unique window into Pliocene-Pleistocene ecosystems, and how they evolved over time as the climate and the landscape changed (*Hopley et al., 2007*; *Pickering et al., 2007*, *2011a*; *Pickering and Kramers, 2010*; *Bailey et al., 2011*; *Reynolds et al., 2011*; *Dirks and Berger, 2013*; *Bruxelles et al., 2014*; *Granger et al., 2015*). For the past 3 Ma, hominin-bearing deposits in caves formed in broadly similar settings involving debris cone accumulations near cave openings (*Partridge and Watt, 1991*; *Brain, 1993*; *Pickering et al., 2007*; *Dirks et al., 2010*; *Bruxelles et al., 2014*). Deposition of these debris cones resulted from granule- to boulder-sized breakdown of ceilings and walls mixed with allochtonous and autochtonous, sand- to clay-sized sediment, as well as peloids and other biogenic debris, by rock falls, debris flow and sheet wash. The clastic sediments vary in texture and composition depending on their position on or near the talus cone, and have been cemented by carbonate-rich waters dripping from cave ceilings.

The sedimentary deposits in the Rising Star cave system that host the remains of the new hominin species of *Homo naledi* (*Berger et al., 2015*) are anomalous when compared to all other deposits of hominin remains in the Cradle of Humankind in a number of significant ways. Not only are the remains extremely numerous and concentrated, but they also occur in largely unconsolidated mud-rich sediments deep inside the cave away from any obvious cave opening, which suggests that a special

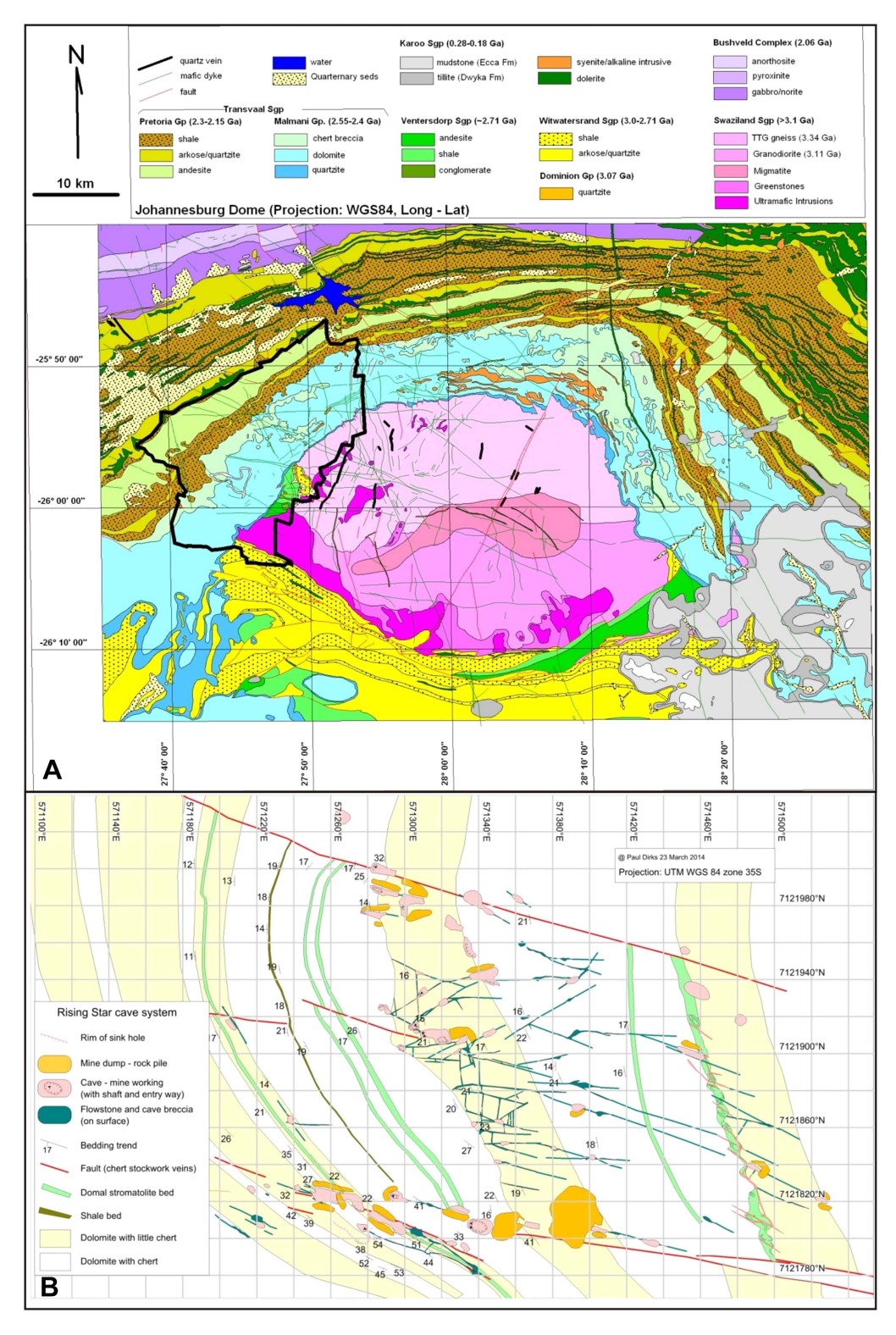

**Figure 1**. Geological setting of Cradle of Humankind and Rising Star cave system. (**A**) Geology of Johannesburg Dome and surroundings, showing the Cradle of Humankind world heritage site in bold black outline. (**B**) surface geology of the immediate surroundings of the Rising Star cave system, showing the fault sets and variable chert content in the dolomite that controlled cave formation. The cave system is confined to a chert-poor stromatolitic dolomite horizon.

confluence of circumstances contributed to their accumulation. This paper provides the geological and taphonomic context to the unusual fossil assemblage of *H. naledi* in the Rising Star cave system, and provides possible scenarios to explain their deposition.

## The Rising Star cave system

The Rising Star cave system lies in the Bloubank River valley, 2.2 km west of Sterkfontein cave. It comprises an area of 250 × 150 m of mapped passageways situated in the core of a gently west dipping (17°) open fold, and is stratigraphically bound to a 15–20 m thick, stromatolitic dolomite horizon in the lower parts of the Monte Christo Formation (*Eriksson et al., 2006*; *Figures 1B and 2A*). This dolomite horizon is largely chert-free, but contains five thin (<10 cm) chert marker horizons that have been used to evaluate the relative position of chambers within the system (*Figure 2B*). The upper contact is marked by a 1–1.3 m-thick, capping chert unit that forms the roof of several large cave chambers. Surface mapping indicates that cave openings and flowstone-filled fractures do not penetrate this capping chert unit, except where a dextral fault truncates the stratigraphy to the south of the system (*Figure 2*). The network of cavities is developed along west-northwest, north and northwest trending fractures and joints.

The fossil-bearing chamber, named the Dinaledi Chamber ('Chamber of Stars' in the Sotho language; *Figure 2A*), is ~30 m below surface and ~80 m, in a straight line, away from the present, nearest entrance to the cave (*Figure 2B*). It is situated in the central part of the system and was found during speleological surveys (see methodology section). The only identified access point into the Dinaledi Chamber involves an exposed, ~15 m climb from the bottom of a large ante-chamber (the Dragon's Back Chamber), up the side of a sharp-edged dolomite block that has dislodged from the roof (the Dragon's Back; *Figure 2B*). From the top of the Dragon's Back, the Dinaledi Chamber is accessed via a narrow, northeast-oriented vertical fissure, and involves a ~12 m vertical climb down, with squeezes as tight as ~20 cm, to reach the floor (*Figure 2B,C*). The main passage forming the Dinaledi Chamber is ~25–50 cm wide at its narrowest and ~10 m long, and expands in width near the intersections with cross-cutting passages (*Figure 2C*). The roof of both the Dinaledi and Dragon's Back chambers is formed by the capping chert (*Figure 2B*). The Dragon's Back Chamber can currently be accessed in two ways, both involving steep climbs along narrow fractures and tight passages (*Figure 2A*): route 1, along an east-northeast trending passage that follows a fracture for a horizontal distance of ~50 m past a narrow access point called the 'postbox'; and route 2, along a more complicated set of broadly east-trending passages, via a network of southeast, east and north trending fractures for ~120 m, and past a narrow access point called 'superman crawl'. Route 1 is the most direct and contains abundant sediment accumulations once the deeper part of the cave is accessed (i.e., ~20 m into the cave); route 2 has a gentler gradient, but is longer and involves a descent along narrow fissures largely devoid of sediment accumulations. An exhaustive search by a professional caving team and researchers has failed to find any other plausible access points into the Dinaledi Chamber, and there is no evidence to suggest that an older, now sealed, entrance to the chamber ever existed. Furthermore, detailed surface mapping of the landscape overlying the Rising Star cave system (*Figure 2A*) illustrates that no large flowstone-filled fractures occur in the region above the Dinaledi Chamber.

The skeletal material recovered from the Rising Star cave was collected during two field expeditions in November 2013 and March 2014, and includes 1550 identifiable fossil hominin specimens as well as six bird and several rodent specimens. Of the 1550 hominin specimens, ~300 numbered bone fragments were collected from the surface of the Dinaledi Chamber, and ~1250 numbered fossil specimens were recovered from a small excavation pit in that chamber.

## Results

### Sedimentology

Throughout the Rising Star cave system erosional remnants of fossiliferous sediment, breccia, and flowstone units provide evidence for several cycles of sediment-flowstone fill and removal/dissolution as the level of the water table in the cave changed repeatedly. On approaching the Dinaledi Chamber from the closest current cave entrance, sediment deposits in the cave become progressively finer-grained. The coarse-grained clastic deposits encountered furthest into the cave are in the Dragon's Back Chamber, and include channelized sandstone and quartz/chert pebble conglomerate units that terminate against the Dragon's Back. The fill generally dips gently west (<5°) in the down slope

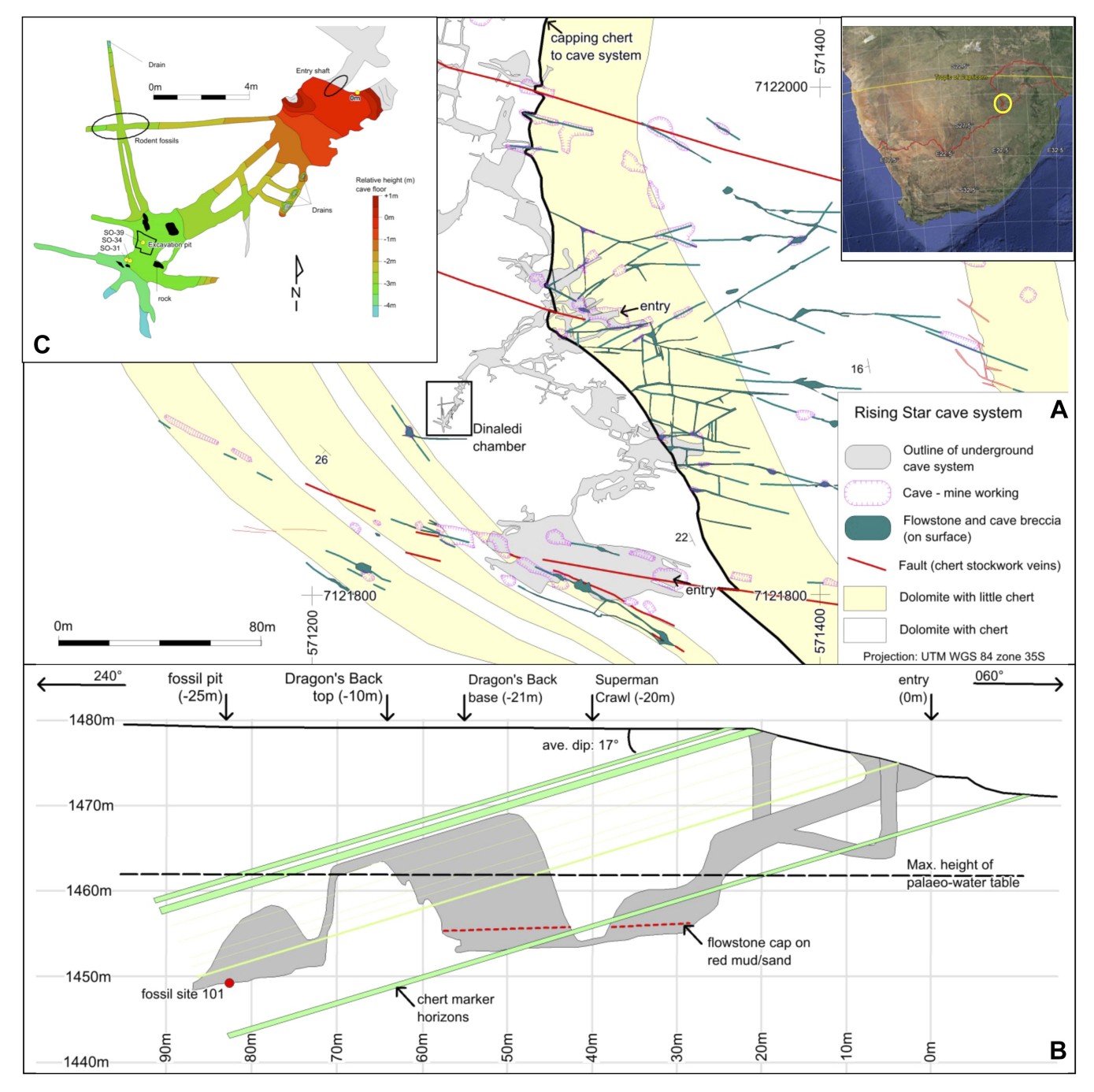

**Figure 2**. Geological map and cross-section of the Rising Star cave system. (**A**) Geological Map showing the distribution of chert-free dolomite and fracture systems controlling the cave. Inset shows the location of the Cradle of Humankind in southern Africa; (**B**) Northeast-Southwest, schematic cross section through the cave system, relative to several chert marker horizons; (**C**) Detailed map of the Dinaledi Chamber showing the orientation of the floor and the position of the excavation and sampling sites.

direction of the passages, and becomes near horizontal in the Dragon's Back and Dinaledi chambers. These chambers contain no evidence of sediment input from proximal sources that would indicate a nearby vertical shaft connecting these chambers to surface (*Pickering et al., 2007*). Mapping reveals no large cave openings on surface above these chambers, although narrow, flowstone-filled fractures

occur (*Figure 2A*) that may have allowed passage of water and some fine-grained sediment into caverns below.

The Dinaledi Chamber is exclusively filled by flowstone and fine-grained sediment involving two depositional facies distributed across three stratigraphic units that filled the chamber over time. Stratigraphic units are separated by erosional unconformities, or laterally continuous flowstone intercalations. Erosion remnants of the units occur in a variety of stratigraphic positions, and there is extensive evidence of reworking with older units being re-deposited into younger units. Each of the facies and units, and each of the flowstone phases that are interlayered with the various units, are described below (*Figures 3, 4*).

## Facies 1: horizontally laminated orange-brown mudstone with sandstone lenses

Facies 1 has been sub-divided into two sub-facies. Facies 1a consists of unlithified, horizontally laminated, orange mud, with very low sand content (*Figure 4A,B*). The composition is dominated by fine sericite clay with subordinate amounts of silt-sized chert and dolomite grains (*Figure 5*). Facies 1a is generally unconsolidated, but contains secondary Mn- and Fe- oxide phases that locally form weakly cemented concretions. Facies 1a has a patchy distribution, occurring both in undisturbed, isolated areas as accumulations atop blocks and in fissures (e.g., *Figure 4A*), and more commonly as erosional remnants of formerly more extensive deposits that filled the Dinaledi Chamber and side passages. Outcrops of Facies 1a (e.g., near the entry point at the top of the chamber, *Figure 4F*) show evidence of in situ auto-brecciation of orange mudstone around exposed margins, due to desiccation, and/or formation of Fe-Mn concretions.

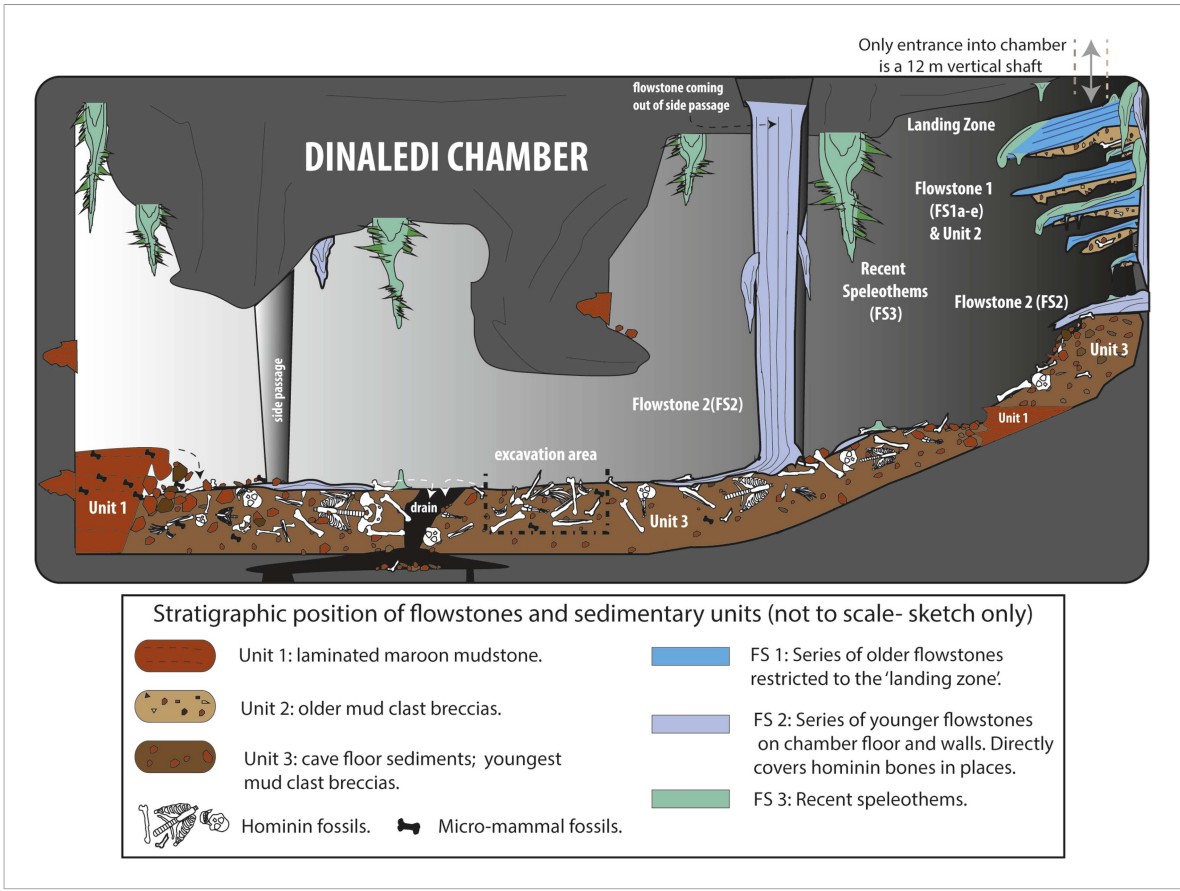

**Figure 3**. Cartoon illustrating the geological and taphonomic context and distribution of fossils, sediments and flowstones within the Dinaledi Chamber. The distribution of the different geological units and flowstones is shown together with the inferred distribution of fossil material.

Facies 1b consists of orange-brown mudstone interlaminated with thin lenses of silt to fine- to medium-grained sand with angular to sub-rounded grains of chert and quartz, and abundant micromammal fossils. The sandstone/siltstone layers are a few mm thick, and in some places display small-scale current ripple laminations. Facies 1b is restricted to isolated erosion remnants in the floor and to crevices in the dolostone above the floor (*Figure 3*). An extensive search for hominin and other macro-vertebrate bones in Facies 1a and 1b was conducted within the Dinaledi Chamber, but none were found.

The horizontally laminated deposits of Facies 1a have been interpreted as autochthonous suspension deposits associated with a low energy environment as water laden with mud gradually filtered into the cave chamber. Deposits of Facies 1a have also accumulated on ledges in isolated fissures where mud filtered through narrow cracks.

Deposits of Facies 1b are interpreted to represent a higher energy environment involving pulses of para-autochthonous and/or allochtonous sediment deposited in the cave by minor sheet flow, which alternated with periods of quiescence during which Facies 1a muds were deposited. The source of micromammal fossils and sand in Facies 1b has not been determined with any certainty, but considering that Facies 1b deposits occur in isolated locations within the chamber, we assume that the sand and fossils may have entered from narrow passageways that drain higher up into the chamber.

## Facies 2: unconsolidated to consolidated, orange-brown mud clast breccia in a mud matrix

Facies 2 consists of massive, that is, non-layered, mud clast breccia surrounded by a matrix of brown mud with localised patches of sparry carbonate cement. Clasts in Facies 2 are predominantly composed of laminated, orange-brown mud fragments (Facies 1a), which are weakly consolidated, angular to sub-angular in shape, and variable in size, but typically <3 cm in diameter (*Figure 4D,E*). Facies 2 locally contains pebble to cobble-sized, angular clasts of dolomite and chert, derived from the surrounding cave walls. No reworked flowstone clasts have been observed in Facies 2 deposits. Sediments of Facies 2 may be clast-supported (e.g., near the landing zone) or matrix supported (e.g., in parts of the cave floor), and the size and angularity of mud clasts varies, as does the degree of matrix and cementation. Macro-vertebrate bones, including all the remains of *H. naledi*, are found within stratigraphic units composed of Facies 2.

Facies 2 deposits are interpreted as accumulations of largely autochthonous debris derived from within the cave chamber through weathering (auto-brecciation) and erosion of Facies 1 deposits, or through reworking of older deposits of Facies 2. The typically angular nature of the soft mud clasts in Facies 2 indicates minimal transport and low-energy processes in the cave chamber. The few fragments of dolomite and chert were sourced from within the chamber, dropped from the cave roofs and walls. Deposits of Facies 2 accumulated in the area beneath the fissure that forms the entry point into the Dinaledi Chamber, to form a debris cone that sloped into the chamber. This debris cone largely developed in the absence of either sustained flowing or standing water to explain its topography and the preservation of angular mud fragments.

### Flowstones in the Dinaledi Chamber

A number of different flowstone formations can be identified in the Dinaledi Chamber. These formed at separate times, and help demarcate stratigraphic units. Each type of flowstone is described below starting with the oldest.

#### Flowstone 1

The oldest flowstones in the cave are a group of five, spatially associated flowstone drapes (Flowstones 1a–e) that appear to be genetically related, and remain as hanging erosion remnants with rounded shapes, near the entrance into the chamber (*Figure 4C*). Remains of Flowstone 1 are preserved to the east of the entry point into the Dinaledi Chamber, where a large, south dipping, dripstone apron (Flowstone 1a) and four partly dissolved, closely spaced flowstone sheets 30–100 cm beneath it (i.e., Flowstones 1b–e) build out from the back of the entry point. All remnants of Flowstone 1 dip ~20–30˚ south towards the bottom of the cave chamber. Flowstone 1a, which is the topmost and thickest (<15 cm) of the early flowstone units covers a 10–15 cm thick remnant of well-indurated Facies 2 sediment assigned to Unit 2 (see below; *Figure 4D*). Flowstones 1b-e cover poorly consolidated remnants of Unit 2 sediment that are partly covered themselves by deposits of younger flowstone

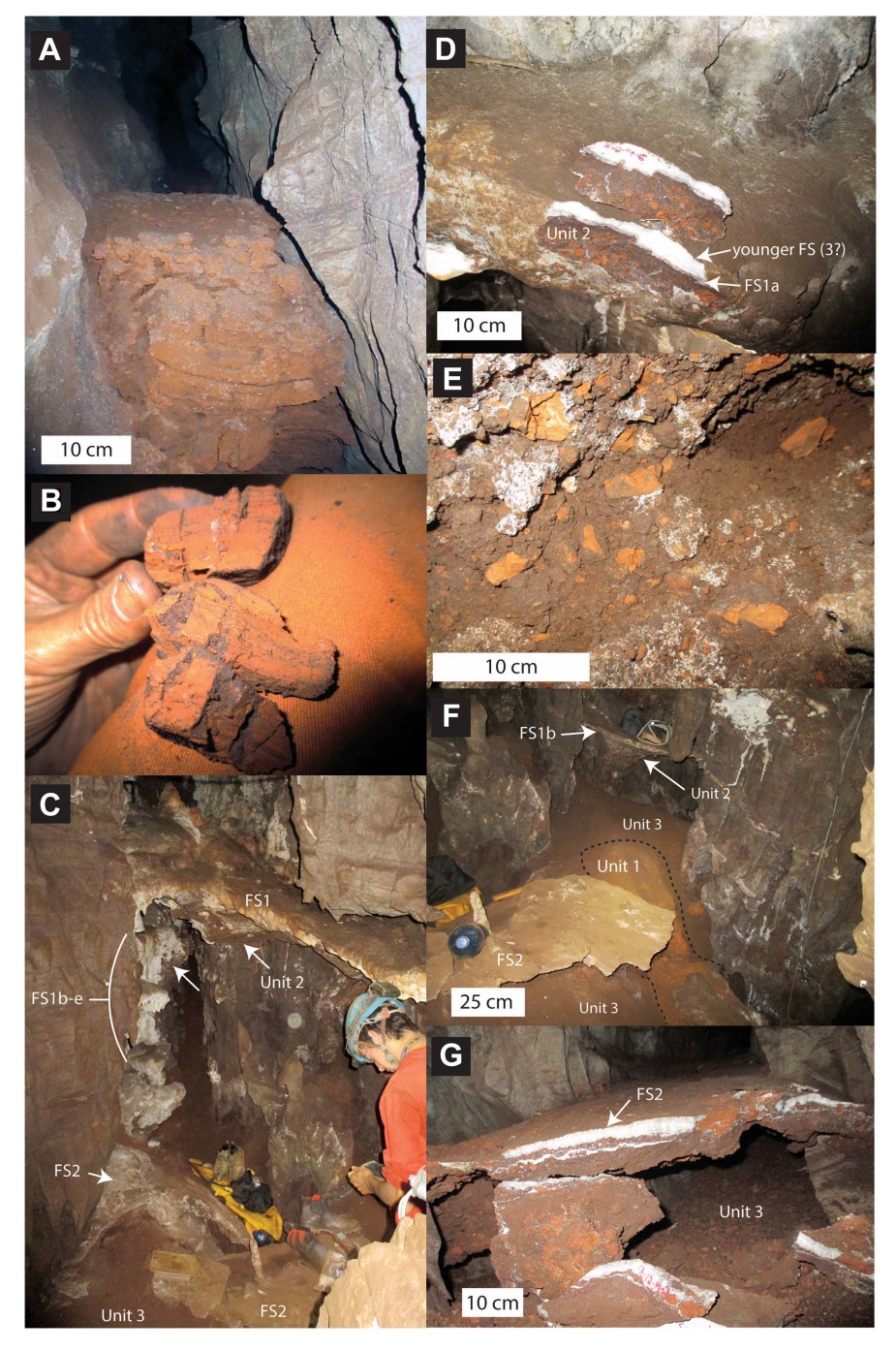

**Figure 4**. Stratigraphic units and flowstones observed in the Dinaledi Chamber. (**A**) Erosional remnant of horizontally laminated Unit 1 strata (Facies 1). (**B**) Close-up view of Unit 1 (Facies 1a) showing fine laminations and small invertebrate burrows (note fine sand infilling in burrows). (**C**) Overview photo of the Dinaledi Chamber, directly to the east of the entrance point into the chamber. Photo shows distribution of Flowstones 1–3 and stratigraphic Units 2 and 3. (**D**) Close-up view of Flowstone 1 encasing sediment of Unit 2. Note that several generations of flowstone (Flowstones 1a–e) are coating Unit 2. The thin, clear lower layer is Flowstone 1a, and the overlying white flowstone is either Flowstone 2 or 3. (**E**) Close-up view of Unit 2, consisting of generally poorly-cemented Facies 2 sediment. (**F**) View of the chamber floor near the entry point. On the cave floor, a large erosional remnant of Unit 1 (orange laminated mudstone of Facies 1a), is surrounded by mud-clast breccia of Unit 3 (main hominin bearing unit). Note that Flowstone 2 has been undercut by post-depositional erosion of Unit 3, which, in this location has resulted in a lowering of the floor by as much as 25 cm. (**G**) Flowstone 2 overlying Unit 3 in one of the chamber's side passages.
*Figure 4. continued on next page*

*Figure 4. Continued*

In this location Unit 3 has also been partly eroded after depositional from underneath the flowstone drape, leaving a hanging remnant, with some indurated sediment of Unit 3 attached to its base. Note the continued deposition of sediment above Flowstone 2.

(i.e., Flowstones 2 or 3). In fresh breaks, Flowstone 1a shows a basal 3–8 mm thick layer of laminated, crystalline calcite, which is covered by an outer layer of mostly white, coarse-crystalline calcite, which varies significantly in thickness and in places forms a fringe of small stalactites along the outer lip of Flowstone 1a (*Figure 4D*). This outer layer likely represents a younger generation of flowstone (Flowstone 3?) that covered Flowstone 1a.

## Flowstone 2

Flowstone 2 represents the most significant phase of flowstone development in the Dinaledi Chamber, and forms cascades emanating from fractures in cave walls and the cave roof throughout the chamber. Deposits of Flowstone 2 partly cover remains of Flowstone 1a–e, and portions of the floor of the cave. This can be observed near the landing zone, but is particularly prominent in the bottom of the chamber. In most side passages, where the dolomite walls narrow to thin fractures, large cascades of Flowstone 2, flow down the cave walls and spread out across the cave floor forming thin flowstone drapes on top of poorly consolidated sediment of Facies 2 containing hominin fossils (Unit 3; *Figure 4F,G*). Locally (e.g., in a side passage on the northwest side of the Dinaledi Chamber) floor drapes of Flowstone 2 cover erosional remnants of Facies 1a (Unit 1). In various places in the chamber, significant undercutting of these floor drapes has occurred due to more recent erosion of the cave floor, creating hanging sheets of Flowstone 2 (*Figure 4F,G*). In fresh breaks, Flowstone 2 varies from white, coarse-crystalline calcite to grey and brown laminated calcite contaminated with mud. Some of the floor drapes have a sugary texture due to post-depositional recrystallisation. In places, Flowstone 2 has been covered by thin deposits of Flowstone 3.

## Flowstone 3

Deposits of Flowstone 3 are currently active flowstone formations in the Dinaledi Chamber, and are primarily restricted to the chamber ceiling in the area around the entrance, where they form delicate needle- and rod-like forms that include well-formed crystals of aragonite. Near the chamber entrance, minor dripstone is also forming on the walls and on top of Flowstone 1 and 2. Lower in the chamber, there are small dripstone accumulations (stalagmites), which have resulted in localised patches of Flowstone 3 on the chamber floor and covering Flowstone 2.

## Stratigraphic units in the Dinaledi Chamber

Using the distribution of flowstones, sedimentary facies (e.g., the presence of Facies 1 mud clasts inside Facies 2), and erosional contacts, a basic stratigraphy of three separate units has been established in the Dinaledi Chamber. This stratigraphic interpretation is preliminary and based solely on geological reasoning (i.e., stratigraphic superposition of individual units, textural variations in Facies 2 sediment and the presence or absence of dissolution features), because of an absence of reliable age data. It is important to note that the facies described above are not synonymous with stratigraphic ordering although in general Facies 1 sediments appear to be older than Facies 2 sediments. Variations in texture, composition and degree of lithification of Facies 2 sediments, as well as direct contact relationships, make it possible to define different stratigraphic units composed of Facies 2. Because we do not yet have a clear understanding of the age relationships, nature of disconformable surfaces or the extent of reworking between the units, we refrain from defining these as discrete allostratigraphic units and instead prefer the use of lithostratigraphic units. The three lithostratigraphic units and their distribution in the Dinaledi Chamber are described below.

## Unit 1

The oldest stratigraphic unit preserved in the Dinaledi Chamber, Unit 1, is represented by outcrops of laminated orange mudstone of Facies 1a, which are preserved as isolated erosion remnants in several places in the Dinaledi Chamber (*Figure 4A,F*). The first significant remnant of Unit 1 is located on the

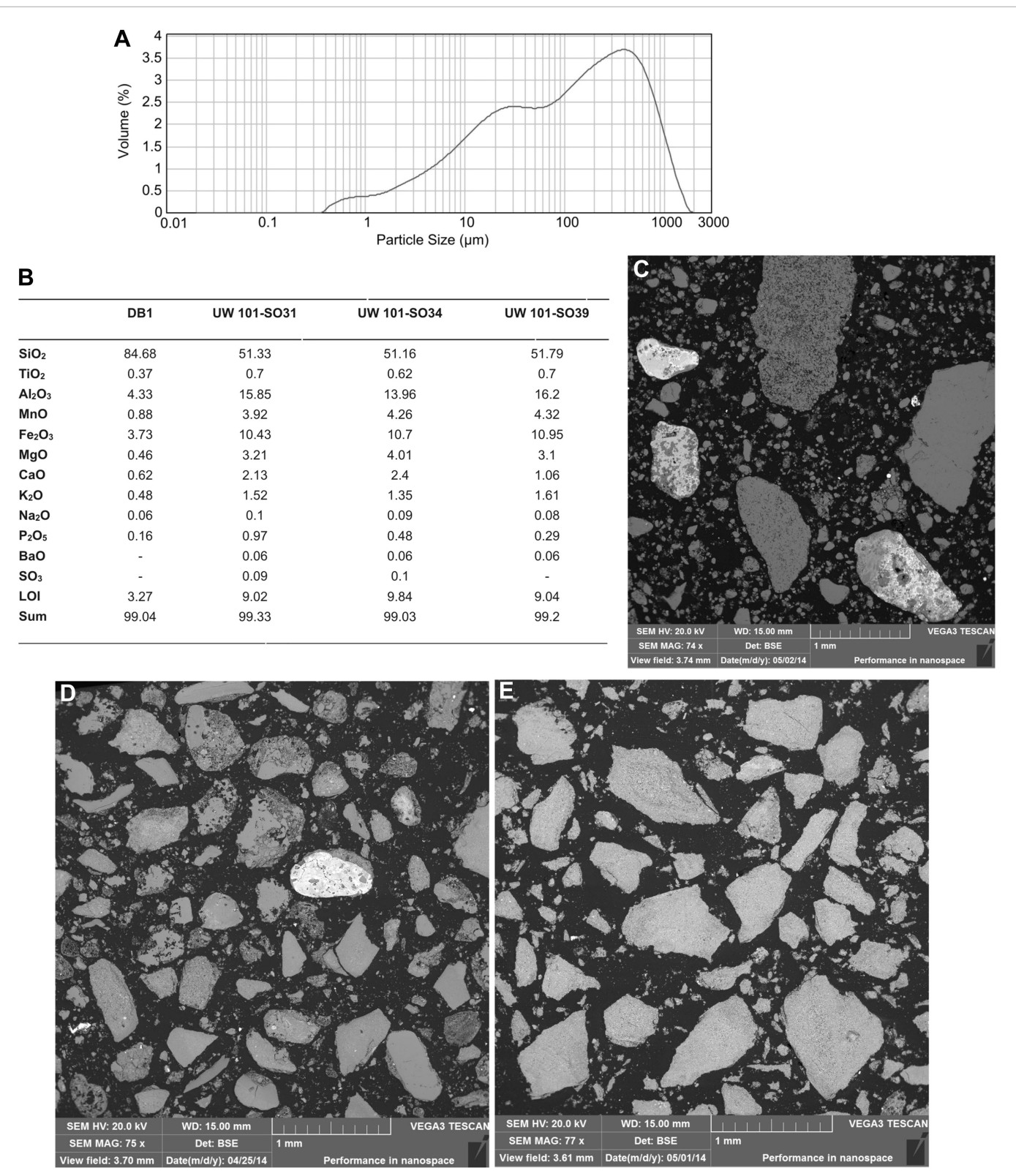

**Figure 5**. Data and characteristics of cave floor sediments (Facies 2) from the Dinaledi and Dragon's Back Chambers. (**A**) Grain size distribution of sample UW101-SO-39 (*Figure 2C*). The bulk of the sample material falls within a size fraction corresponding to silt and fine-grained sand. Some coarser mudstone fragments did not disintegrate when immersed in water, likely due to considerable Mn- and Fe-oxide micro-concretionary development in the orange

*Figure 5. continued on next page*

*Figure 5. Continued*

mudstone. Because some mudstone fragments are well lithified the particle size distribution is skewed towards the coarser grain-size values. (**B**) Results of XRF analyses of bulk samples of three floor sediments from the Dinaledi Chamber (UW101-SO31, -34 and -39) and one from the Dragon's Back chamber (DB-1). The sample from the Dragon's Back Chamber has a radically different composition from those of the Dinaledi Chamber, with the high $SiO_2$ content reflecting its dominance of quartz. The Dinaledi samples have much higher $Al_2O_3$ and $K_2O$ contents than DB-1, indicating a higher content of clay minerals and mica, and higher CaO, MgO, MnO, and total Fe oxide contents which reflect alterations and inclusions. The higher $P_2O_5$ content of the Dinaledi samples is probably located in comminuted bone fragments which are seen macroscopically. The volatiles content (LOI) of the Dinaledi samples is also higher than in DB-1, in accord with a higher total clay mineral and mica content. (**C–E**) Backscattered electron (BSE) wide-field images of grain mounts from floor sediments. Brighter shades indicate the presence of heavier elements, mainly Mn and Fe in altered grains. (**C**) DB-1, Dragon's Back Chamber, large fragments are quartz and chert, partly altered. (**D**) UW101-SO34. (**E**) UW101-SO39. In these samples the large fragments are almost exclusively clay; note their angular shape which shows these to be locally derived.

floor of the cave chamber, to the east of the entry point below outcrops of younger units composed of Facies 2, and remnants of Flowstones 1–3. This exposure of Unit 1 is largely buried by younger sediment belonging to Unit 3 (*Figure 4F*).

A second large outcrop of Unit 1 occurs in a northwest-trending side passage of the Dinaledi Chamber, where it forms a ~4 m long portion of cave floor composed of finely laminated orange mud. At some point in time, deposits of Unit 1 would have filled much of the Dinaledi Chamber, and erosion products (i.e., mud clasts) from this unit are the source for much of the mud clast breccia (i.e., deposits of Facies 2; *Figure 4E*) observed in the cave chamber.

The isolated outcrop of micromammal-bearing, orange mudstone-sandstone (Facies 1b) in one of the side passages may represent a further outcrop of Unit 1, in which case it provides evidence for lateral facies variations in Unit 1. Alternatively the outcrop of Facies 1b represents local reworking of Unit 1, and hence a younger stratigraphic unit. Because of this uncertainly, we presently do not include Facies 1b within Unit 1.

Small pockets of Facies 1a sediment (a few tens of cm long and wide by up to 10 cm thick) are preserved in crevices and as hanging remnants, high-up in the Dinaledi Chamber, up to 4 m above the chamber floor. These outcrops may represent isolated erosion remnants of Unit 1, but in the absence of direct stratigraphic relationships we cannot be sure. Alternatively, Facies 1a has a long depositional history, and these isolated outcrops represent in situ accumulations of 'wad' formed as a result of weathering of the dolomite wall rock (*Martini et al., 2003*).

## Unit 2

The next stratigraphic unit, Unit 2, is the oldest unit composed of Facies 2 sediment, and is restricted to small outcrops below the entrance shaft into the Dinaledi Chamber. Unit 2 is a composite unit that consists of variably consolidated, clast-supported Facies 2 sediment, dominated by reworked mud clasts derived from Unit 1, mixed with rare chert and dolomite fragments (*Figure 4D*). The mud clasts in this unit are composed of a combination of fresh (orange), angular clasts, and more oxidized (brown), sub-angular to sub-rounded clasts (*Figure 4E*), set in an oxidized, brown muddy matrix with some silt- to sand-sized particles, and locally cemented by sparry carbonate.

Outcrops of Unit 2 are preserved as erosional remnants of limited size that are attached below the hanging aprons of Flowstones 1a–e. The topmost outcrop of Unit 2 occurs below Flowstone 1a, and consists of a 10–15 cm thick erosion remnant of consolidated mud clast breccia (*Figure 4D*). The outer surface of this outcrop has a smooth appearance as a result of dissolution after deposition and lithification. In contrast, erosion remnants of Unit 2 that are preserved underneath aprons of Flowstones 1b–e (*Figure 4C*), are mostly unconsolidated and have a rubbly appearance.

Like the overlying flowstone sheets, outcrops of Unit 2 dip southwest at ~20–30° into the Dinaledi Chamber (*Figure 4C*), which indicates that Unit 2 accumulated on a debris cone or rubble deposit that sloped into the chamber and developed below the entry shaft. Sediments that accumulated on this debris cone gradually slumped into the Dinaledi Chamber at a time that flowstone was forming, resulting in the progressive formation of Flowstones 1a–e. This process of progressive erosion is described in more detail in the discussion section.

Outcrops of Unit 2 contain several, in situ macro-fossil bones that can be identified as hominin, but are otherwise undiagnostic. This includes the shafts of a juvenile hominin ulna and radius contained in

a collapsed portion of unconsolidated Unit 2 sediment derived from below Flowstone 1b, which are consistent with the remains of *H. naledi*.

## Unit 3

Unit 3 is the youngest stratigraphic unit in the Dinaledi Chamber, and is represented by sediment that accumulated along the floor of the chamber and along the bottom of many, but not all of the side passages (*Figure 4G*). Unit 3 is composed of largely unconsolidated sediment of Facies 2 dominated by reworked orange mud clasts embedded in a brown muddy matrix. This unit is massive and displays no evidence of layering or grading, and is commonly matrix supported (as opposed to the mostly clast-supported Unit 2). It has been derived from weathering and erosion of Units 1 and 2. Unit 3 partly covers exposures of Unit 1 in several side passages and near the entry point into the cave.

Upon initial discovery of the Dinaledi Chamber, Unit 3 contained dozens of hominin bones exposed along its surface and partially buried within it, in most portions of the chamber where Unit 3 has formed (*Figure 6*). A series of minor test pits (a few cm deep) were dug into Unit 3 at various areas of the cave floor, which revealed that abundant additional hominin bones are buried at shallow depths within Unit 3 throughout much of the chamber. The abundance and density of *H. naledi* bones in the chamber is demonstrated in the single excavation, where most (1250 out of 1550 elements) of the fossil material documented from the Dinaledi Chamber was collected (*Figure 7*). Unit 3 also contains rare disarticulated rodent remains that are undiagnostic of age, and that are possibly derived as an erosional product from Unit 1 (Facies 1b) or that were deposited directly into Unit 3 as it accumulated.

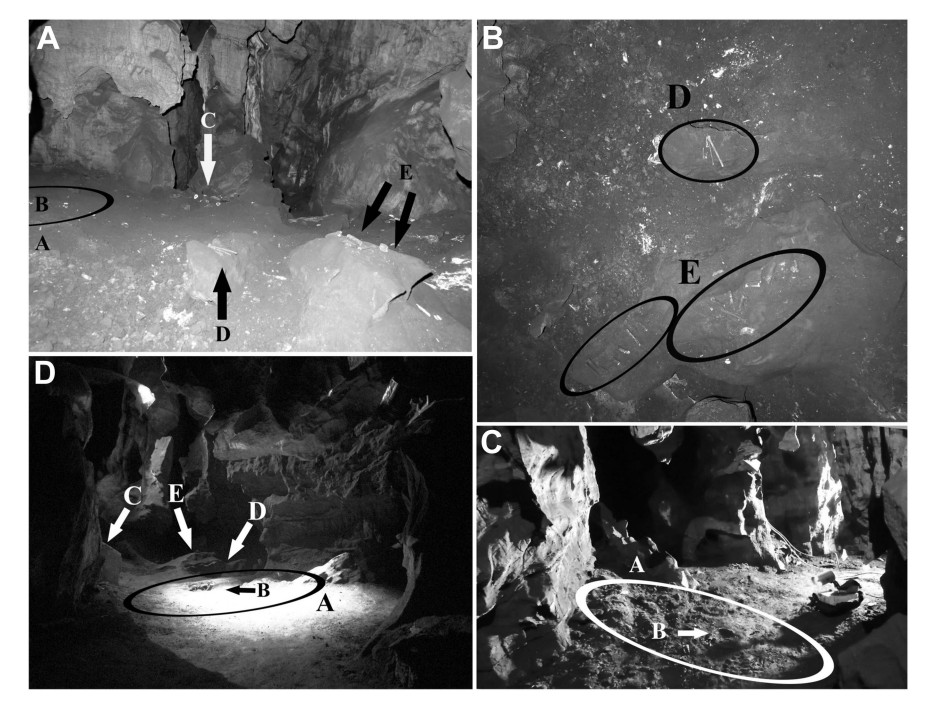

**Figure 6**. Views of the Dinaledi Chamber. Clockwise from top left: (**A**) Photograph taken during initial exploration of the chamber; view to the N. Ellipse 'A' indicates the area where most of the hominid material was excavated. Letter 'B' shows the location of the cranial fragment that was one of the first pieces removed from the chamber. Arrows 'C, D and E' indicate areas of concentrated surface material. Block E is ∼ 50 cm across. (**B**) pre-excavation view of the chamber floor. 'D' and 'E' included hominid teeth, (intrusive) bird bones and several long bone fragments that had been 'arranged' on rocks by an unknown caver prior to discovery by our caving team. Top of the photograph point to the NE. Base of the photograph is 80 cm. (**C**) view of the primary excavation area prior to excavation looking in a NE direction. The diameter of the ellipse is 1 m for scale (**D**) the excavation area at the end of the first round of excavations looking in a SSE direction (November 2013). Ellipse A is the same as in previous figures. Arrow B shows the fragment of a long bone that was uncovered after the cranial fragment had been removed. Arrows C, D and E indicate where surface concentrations had previously been. The diameter of the ellipse is 1 m for scale.

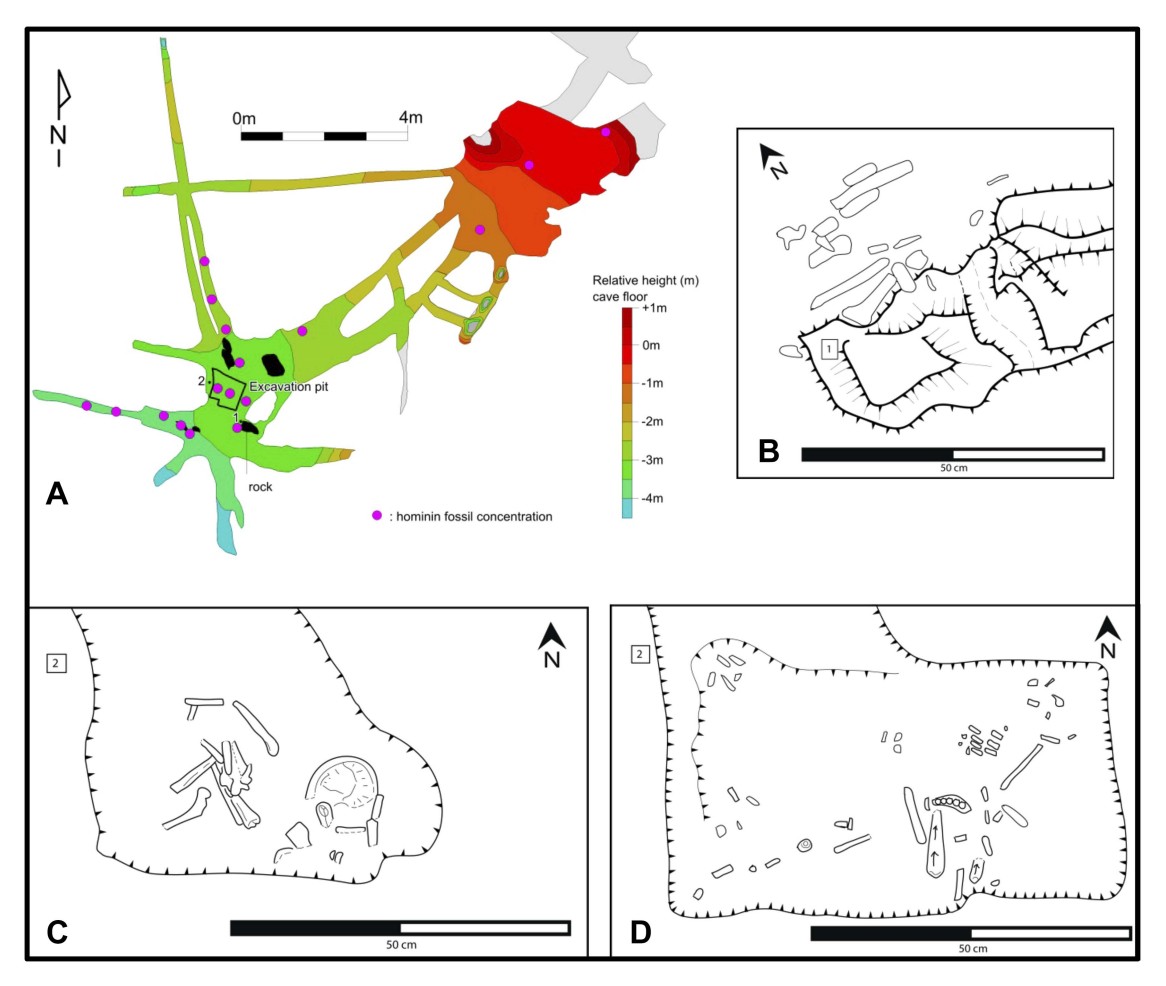

**Figure 7**. Map of the cave chamber showing the distribution of hominin fossils. (**A**) distribution of concentrations of bone fragments of *H. naledi* along the floor of the Dinaledi Chamber. The positions of maps (**B**) and (**C**), (**D**) are shown relative to survey pegs 1 and 2 respectively. (**B**) Concentration of long bone fragments encountered next to a rock embedded in Unit 3 sediment. (**C**) Distribution of fossils in the excavation pit at the start of the excavations in November 2013 (~5 cm below surface). (**D**) Distribution of fossils in the excavation pit during excavations in March 2014 (~15 cm below surface); the long-bone in the central part of the pit is in a near-vertical position and is shown in *Figure 9C*.

In many parts of the chamber, Unit 3 is covered by a thin sheet of Flowstone 2, which spread across the top of Unit 3 from a number of separate drip points. When dated, these cover sheets of Flowstone 2 will constrain the youngest possible age for the *H. Naledi* fossil assemblage. The formation of drapes of Flowstone 2 atop Unit 3 was followed by localized, post-depositional erosion of the cave floor, a process that continues today, as sediment is periodically washed down floor drains in the chamber. As a result, hanging erosion remnants of Flowstone 2 are preserved in several places along the walls of the cave chamber (e.g., directly north of the excavation pit), where they remain up to 20 cm above the present cave floor (*Figure 4F,G*).

## Mineralogy and chemistry

Mineralogical and chemical studies were carried out on sediment samples from the floors of the Dinaledi and Dragon's Back chambers (Facies 2; Unit 3) to assess the nature of the sediment encasing the Dinaledi fossils and to determine whether these chambers were connected at the time of deposition of Units 2 and 3 (see *Supplementary file 1* for analytical results).

Four samples of cave sediment were studied using different techniques to obtain data on their size fraction, mineralogy and geochemical characteristics. Samples UW101-SO-31, UW101-SO-34 and

UW101-SO-39 were collected near the excavation pit in the Dinaledi Chamber, from brown matrix-supported mud clast breccia of Unit 3 (*Figures 2C and 7*). UW101-SO-31 is light-brown in colour, and contains sand grains and has bone fragments. UW101-SO-34 consists of dark-brown mud and is not gritty to feel. UW101-SO-39 is a medium- to dark-brown mud that is slightly gritty to feel. One additional sample, DB-1, was collected for analysis from unconsolidated floor sediment in the Dragon's Back chamber (*Figure 2B*). This is a brown silty mud with a fine-gritty feel, containing sub-mm sized bone fragments.

The three samples of Unit 3 from the Dinaledi Chamber (SO31, 34, 39; *Figures 2C and 5D,E, 8*) are dominated by reworked, angular mud clasts in a clay matrix, with some chert fragments, but with little externally derived detrital quartz. The sample from the Dragon's Back Chamber is dominated by detrital quartz, with some detrital muscovite, as well as shale and chert fragments, of which some are altered and impregnated and/or coated with Mn- and Fe-oxides. Mudstone fragments are rare in DB-1 (*Figures 5C and 8*). Elevated MnO (3.9–4.2%) and $Fe_2O_3$ (10.3–11%) levels in the floor sediments of both chambers are associated with alteration of chert or mudstone prior to their comminution, with Mn- and Fe oxides/hydroxides occurring as replacement and micro-vein infilling. XRD diffractograms identified both quartz and muscovite in all samples with high scores compared to other minerals. Hematite was identified with high scores in all the Dinaledi samples. Goethite ($FeO(OH)$) and birnessite ($(Na,Ca,K)(Mn^{4+},Mn^{3+})_2O_4 \cdot 1.5H_2O$) were only identified in UW101-SO-31, with low scores, but high certainty. Other minerals identified with low scores and high certainties are dolomite ($(CaMg)(CO_3)_2$) in UW101-SO-34 and kaolinite ($Al2Si_2O_5(OH)_4$) in UW101-SO-39. Using XRD, only quartz and muscovite were identified in DB-1. Acid tests showed all samples to be free of calcite or aragonite. All analysed Mn oxi-hydroxides in both chambers contain fluorine, with similar (atomic) F/Mn ratios of $\approx 0.14$, indicating a similar chemical environment during their formation. Different patterns in $K_2O$ vs $Al_2O_3$ plots reflect a dominance of mudstone fragments consisting mainly of clay minerals, in the Dinaledi Chamber, and the presence of muscovite in the Dragon's Back Chamber (*Figure 8*). The contrasting composition in particulate matter of floor sediments in the two chambers, suggests that the Dinaledi Chamber was an isolated sedimentary environment at the time of deposition of Unit 3, with no or very limited transfer of sediment between the two chambers.

Flowstone samples from the Dinaledi Chamber were analysed for uranium to assess the possibility of U-Pb dating. Although analysed samples mostly contain sufficient U for this (0.3–0.7 ppm), a fine dusting of a detrital component derived from associated muds is present in all tested pilot samples. This has confounded preliminary attempts at U-Pb dating, because of the high, and isotopically variable, background of common Pb it carries.

## Taphonomy

A detailed taphonomic study of the hominin remains is ongoing, and initial results have been provided in this study to illustrate the broader context of the fossil assemblage. Analyses of taphonomic processes affecting the Dinaledi hominin fossil assemblage have been conducted to describe decomposition, weathering and fracture patterns, surface modifications as a result of invertebrate–bone interactions (as well as the lack of vertebrate–bone interactions), and spatial context including skeletal distribution patterns (*Behrensmeyer et al., 1986*; *Galloway, 1999*; *Straus and Porada, 2003*; *Loe, 2009*; *Symes et al., 2013*; *Figures 7, 9–12*). Summary tables of results are presented in *Table 1* and *Supplementary file 2*.

### Bone material and its spatial distribution

*Figure 7* shows the general distribution of fossil material within the Dinaledi Chamber, together with more detailed maps of fossil distributions in the excavation pit and nearby surficial areas. When the Dinaledi Chamber was first entered, the sediments along the cave floor (i.e., Unit 3) consisted largely of loosely packed, semi-moist, clay-rich clumps of varying sizes in which bone material was distributed. Where people had moved through the chamber, the sediment along the floor had been compacted down to a flat, semi-hard surface. The hominin bone material was distributed in Unit 3, across the surface in almost every area of the chamber, including narrow side passages and offshoots, with the highest concentration of bone material encountered near the southwest end of the chamber, about 10–12 m downslope from the entry point, where the floor levels out.

Hominin bones were found at the surface of, and within Unit 3, distributed across the floor of most of the Dinaledi Chamber, and short distances down many side passages (*Figures 6, 7*). In parts of the

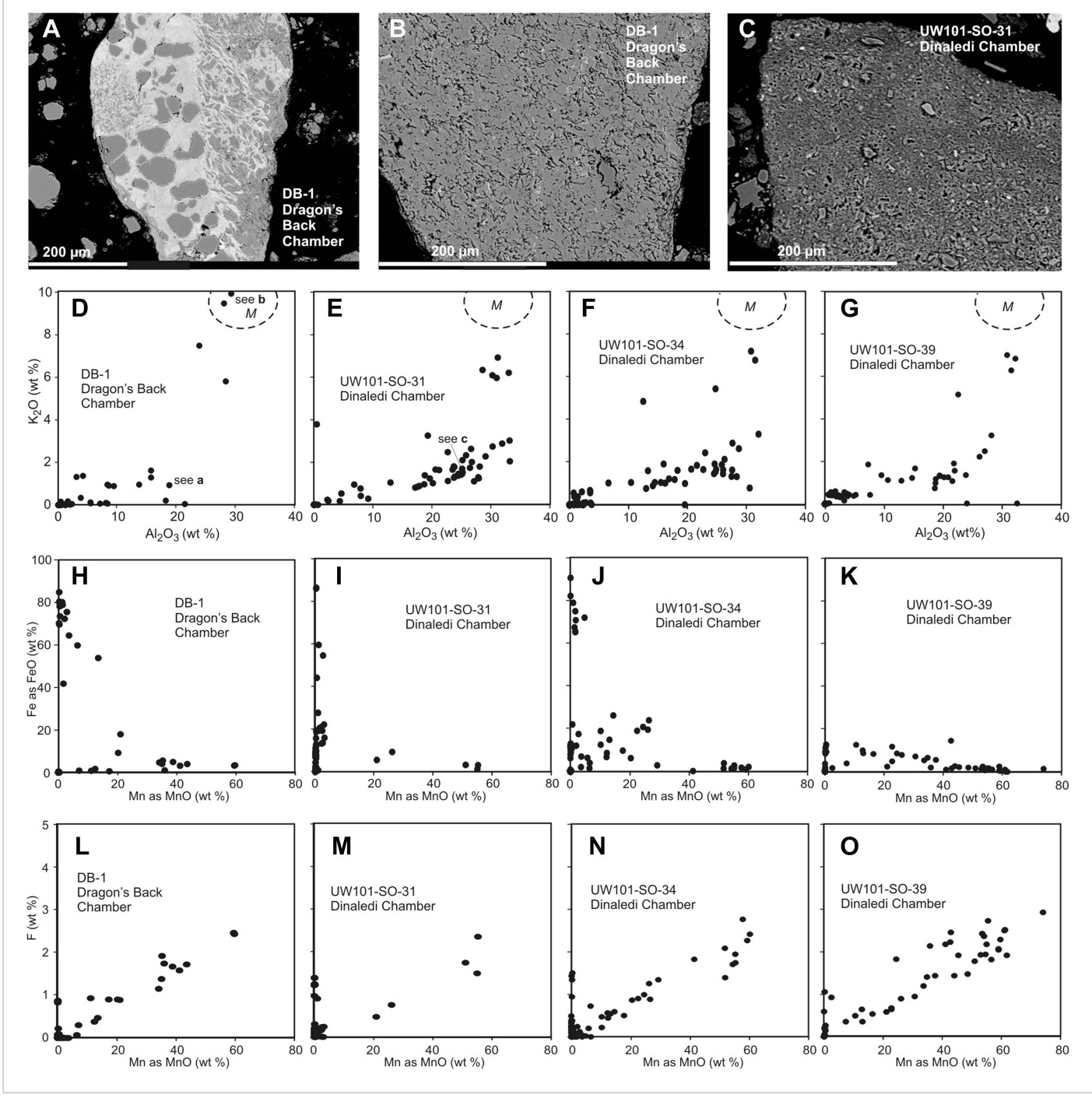

**Figure 8.** Comparison of selected fragments and electron microprobe analytical data of Facies 2 (Unit 3, floor) sediment in the Dinaledi Chamber, and floor sediments in the Dragon's Back Chamber. Analytical spot size is 5 μm diameter, which is generally larger than grain sizes. (**A**) Chert fragment impregnated with Mn oxi-hydroxide from the Dragon's Back Chamber. (**B**) Shale fragment from the Dragon's Back Chamber. (**C**) orange mud clast, typical of Facies 2 sediments from the Dinaledi Chamber, note much finer grain size than seen in (**B**). (**D–G**) Plots of $K_2O$ vs $Al_2O_3$ for mud clast fragments in Facies 2 samples from both chambers show an important difference between them. *M*, muscovite compositional field. The samples from the Dinaledi Chamber (**E–G**) yield some data close to the muscovite field, probably indicating sericite grains slightly smaller than the spot size, and all show a trend with K/Al ratios much lower than muscovite, up to a high $Al_2O_3$ content >30%, which indicates either illite, or mixtures of sericite and kaolinite or other K-free clay minerals. In (**E**) the analysis of the fragment shown in (**C**) is indicated. The sample from the Dragon's Back Chamber in (**D**) shows data in the muscovite field (data point corresponding to [**B**] is indicated in [**D**]), but otherwise only low K- and Al-concentrations, which are typical of Mn oxi-hydroxide impregnation (data point corresponding to [a] is indicated in [**D**]). No analytical data in d correspond to mudstone fragments such as shown in (**C**). (**H–K**), plots of Fe as

*Figure 8. continued on next page*

Figure 8. Continued

FeO vs Mn as MnO show similarity between the chambers with respect to Fe-Mn oxi-hydroxide impregnations and alterations within the fragments: in both cases, domains with high Fe rarely coincide with domains high in Mn. (**L–O**), Plots of F vs Mn as MnO. Some elevated F concentrations at zero Mn values occur, but most data show a correlation for samples from both chambers: elevated Mn content is invariably associated with elevated F, with an atomic ratio F/Mn ≈ 0.14. No Mn oxi-hydroxide minerals with F have been described, but partial substitution of F⁻ for OH⁻ as in apatite might be suspected. The similarity of the F/Mn ratios in Mn oxi-hydroxide impregnated fragments from Facies 2 sediments in both chambers suggests a uniform geochemical environment during the Mn oxi-hydroxide alteration event.

chamber, Flowstone 2 covers Unit 3 along the surface of the cave floor, encasing some hominin bones. A few hominin bones were also collected from loose sediment accumulated on top of Flowstone 2, and represent re-deposited material derived from recently eroded and transported deposits of Unit 3. In situ bone fragments exist in Unit 2, which are hominin and contributed to the assemblage found in Unit 3.

To date, most hominin bones have been collected from Unit 3. A single excavation of Unit 3 was conducted in a 0.8 × 0.8 m pit, ~20–25 cm deep (with a 50 cm deep central sondage; *Figure 2C*), near

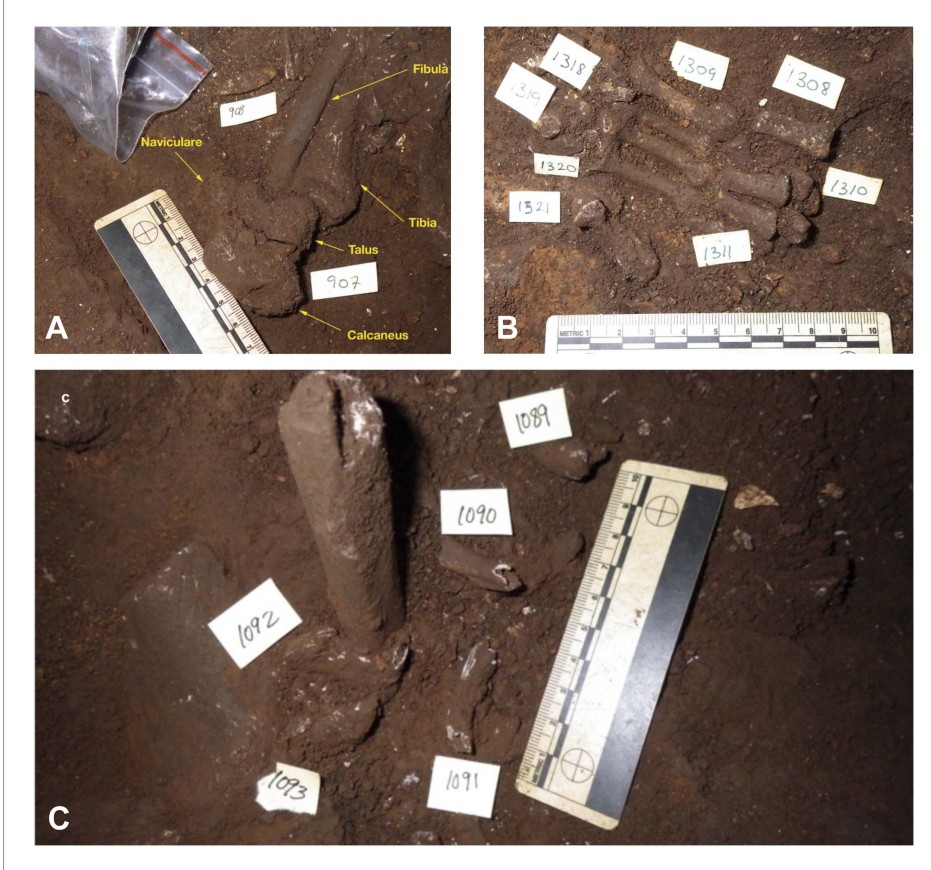

**Figure 9**. Taphonomic spatial patterning within the fossil assemblage exposed in the excavation pit. Taphonomic signatures and spatial orientations suggest that some of the assemblage may be para-authochthonous in nature, rather than primary or in situ. This scenario provides a mechanism for explaining the combination of near- or fully-anatomically articulated skeletal material and elements, which are heavily commingled and in a non-horizontal resting state (from near-vertical to oblique long-axis orientations). (**A**) Example of an articulated ankle region. (**B**) Example of an articulated hand. (**C**) Example of cluster of skeletal elements showing disarticulated elements in a non-horizontal resting state. Note long bone fragment in near-vertical alignment, compared to normal horizontal or near-horizontal alignment of the commingled elements surrounding it. Labels denote specimen numbers.

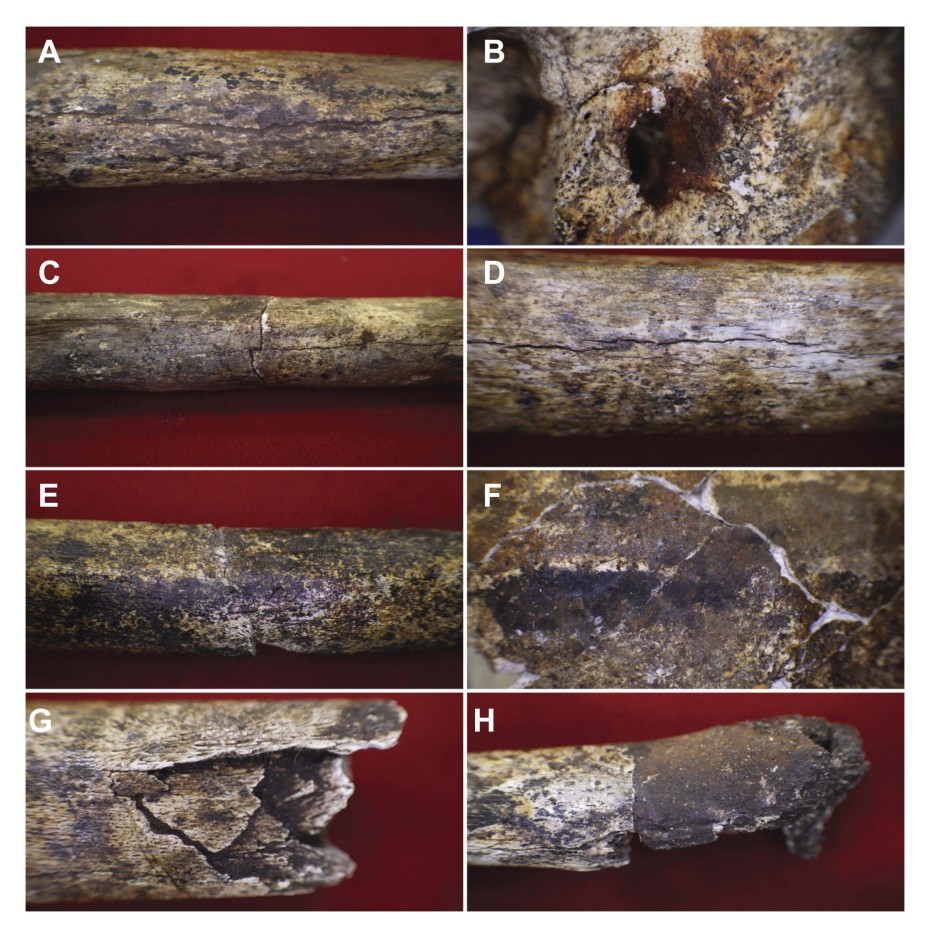

**Figure 10**. Examples of taphonomic traces recorded on hominin remains. (**A**) UW101–1288 tibial diaphysis showing evidence of mineral staining adhering to the cortex. The fossil shows evidence of dark zone sub-aerial or sub-surface weathering. Specimen shows a central midline crack with sediment infill, which separates conjoined manganese concretions. (**B**) UW101–419 (Cranium A[1]) showing iron oxide staining around the external auditory meatus. (**C**) UW101–312 and 1040 conjoined fragments of a tibial shaft, showing stepped transverse fracture (post-mortem) of the mid-shaft; note longitudinal crack, and evidence of invertebrate modification. (**D**) UW101–1288 tibial diaphysis showing a weathering pattern typical of Stage 1 evidenced by fine longitudinal cracks, without concomitant flaking, delamination, or the formation of fibrous texture. (**E**) UW101–1074 tibial shaft showing manganese mineral concretions overlying yellow staining across the diaphysis. (**F**) Specimen UW 101–419 Cranium A(1) displaying tide lines of dark brown, reddish brown and yellow staining, which extends across different vault fragment. (**G**) UW101–498 tibial shaft, showing comminuted post-mortem fracture/crushing preserved by sediment infiltrate. (**H**) UW101–1070 segment of tibial diaphysis displaying differential mineral staining patterns between conjoined fragments.

a dense surface accumulation of bone where the floor levels out and the chamber widens (*Figure 6C,D, 7*). The approximately 1250 specimens recovered from this excavation consist of a variety of articulated, disarticulated and commingled cranial and postcranial material (*Figure 7*). Articulated elements appear more common in the lower 10 cm of the excavation and include critical elements such as a hand, ankle and foot (*Berger et al., 2015*; *Figure 9*). At ~15 cm depth, skeletal material was found less frequently, and after reaching a depth of 20 cm, no further fossil remains were recovered. It is not presently known whether fossil concentrations and depths are consistent through the remainder of the chamber.

Within the excavation pit the recovered elements present a mixture of skeletal material in near-primary (autochthonous) and secondary (re-deposited) context (*Figure 9*). The excavation revealed

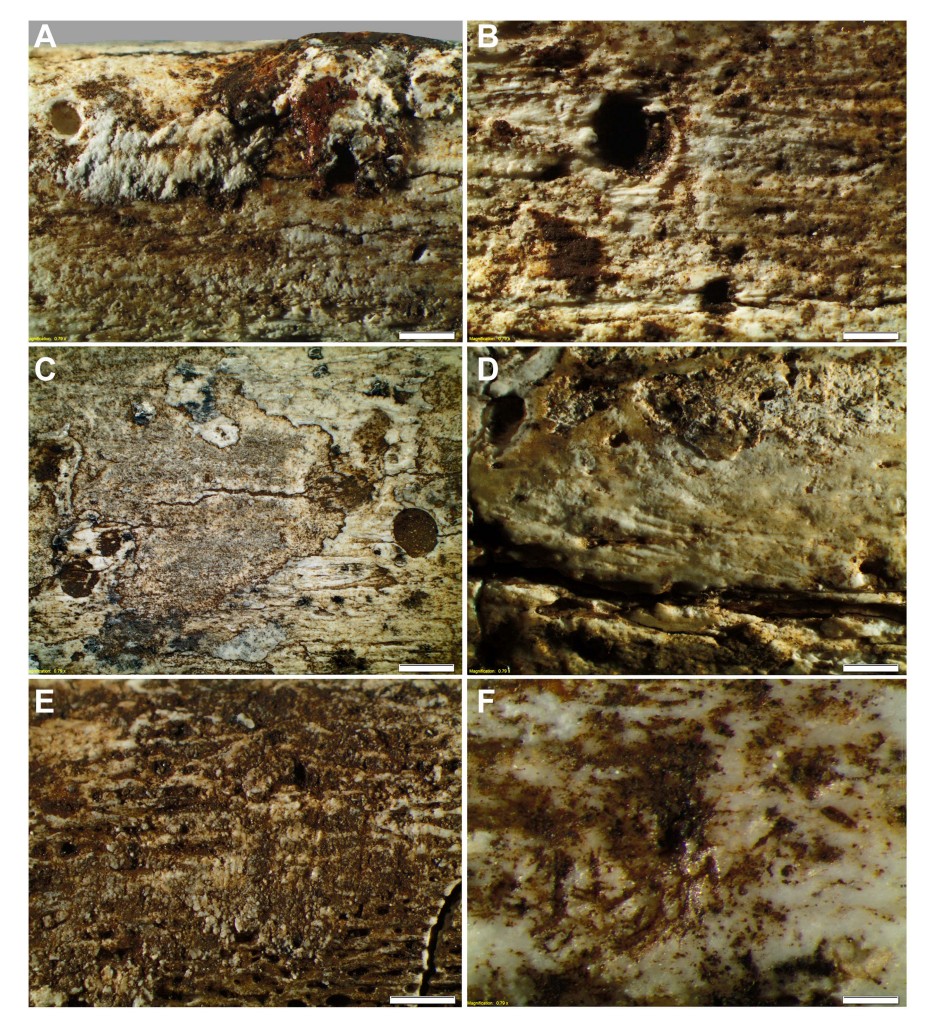

**Figure 11**. Taphonomy—surface modifications. (**A**) Removal of the bone surface with sets of shallow, evenly spaced, multiple parallel striations on fibula (UW101–1037), which run longitudinal with the main axis of the bone and are interpreted as gastropod radula damage. (**B**) Fibula (UW101–1037) showing removal of the bone surface with sets of shallow, evenly spaced, multiple parallel striations that follow the collagen fibres together with shallow circular pits ranging from 0.1 to 3 mm in diameter, the bases of which may be smooth, cupped, or covered with multiple parallel striations. These features have been attributed to gastropod radula damage. (**C**) Tibia (UW101–484) showing removal of the bone surface with sets of shallow, striations that show a smooth scalloped edge together with circular pits ranging from 0.1 to 3 mm in diameter interpreted as the result of gnawing by beetle larvae. (**D**) Tibia (UW101–484) with areas of surface removal that have a straight edge associated with scrape marks interpreted as damage made by a beetle mandible. (**E**) Fibula (UW101–1037) with sets of shallow, evenly spaced, multiple parallel striations orientated transverse to the long axis of the bone interpreted as gastropod radula damage, resulting in an etched surface appearance that exposes underlying structures. (**F**) Tibia (UW101–484) showing clusters of large individual striations that are variably arrow-shaped and often overlap, interpreted as damage made by a beetle mandible. Compare with *Figure 12* which shows surface modifications made by modern snails and beetles and their larvae. The scale bar in all samples equals 1 mm.

a range of long bones, teeth and various other skeletal fragments as well as a nearly complete, but highly fragmented cranium at the upper levels of the deposit, immediately underlain by a nearly complete and articulated lower limb of a child (including articulated tibia, fibula and talus), in turn underlain by a fully articulated hand of an adult. Within the pit the vast majority of long bones were found in horizontal to near-horizontal positions, with limited orientational alignment. The horizontal bones were found interspersed with a small number of long bones positioned in a non-horizontal

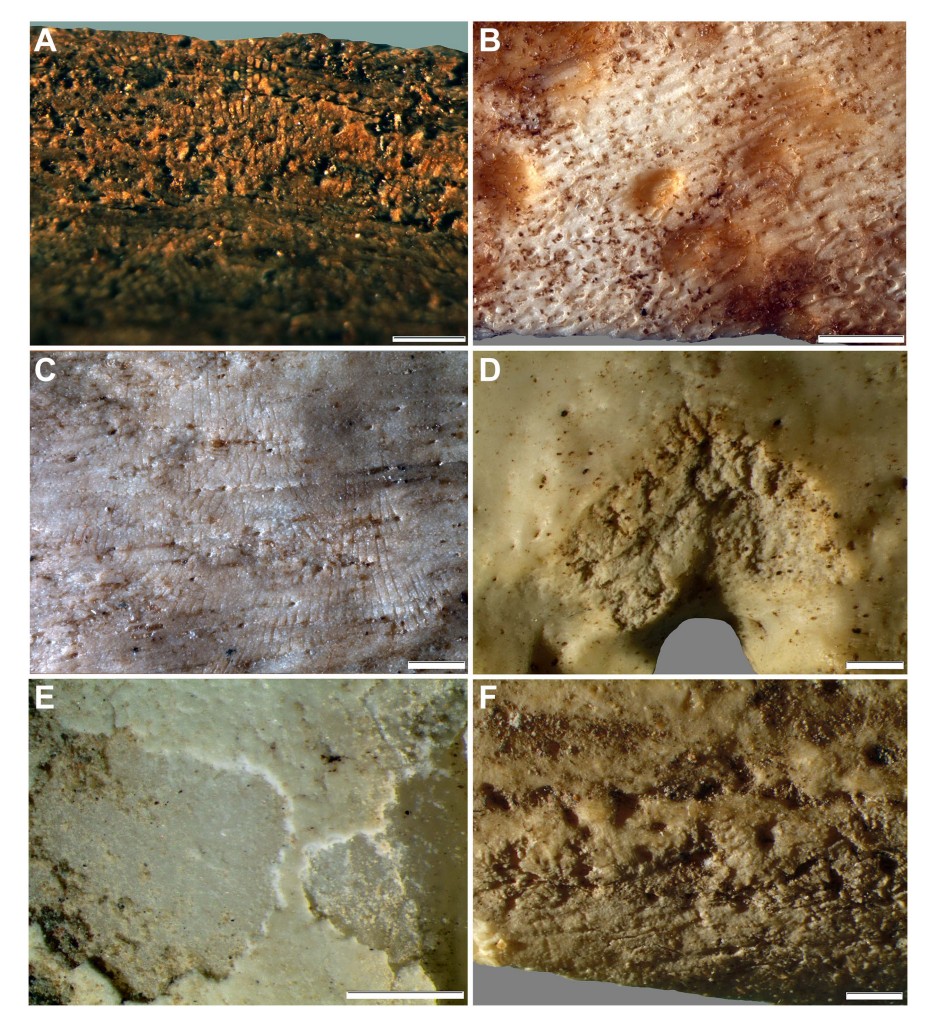

**Figure 12**. Comparative examples of surface modifications on bone made by modern snails and beetles and their larvae after four months in controlled experiments. Gastropods and beetles were found to produce similar modifications to those observed on the Rising Star hominin remains, and remove the surfaces of fresh, dry and fossil bones to an equal degree (see *Figure 11*). (**A**) Dry bovid rib showing surface removal associated with evenly spaced, multiple parallel striations made by the radula of an *Achatina* (land snail). (**B**) Fresh sheep bone that was originally covered with tissue showing how *Helix aspersa* (garden snails) have removed the outer cortical lamellae to produce an etched appearance and create circular shallow pits with smooth and striated bases. (**C**) Dry bovid rib showing shallow, evenly spaced, multiple parallel striations produced by *Achatina*. (**D**) Dry bird femur showing large individual striations that are variably arrow-shaped and often overlap, made by *Omorgus squalidus* (hide beetles). (**E**) A weathered bovid tooth showing surface removal with a scalloped edge produced by *Dermestes maculatus* larvae, and with a straight edge associated with scrape marks. (**F**) Scrape marks created by a *D. maculatus* adult beetle mandible on a dry medium-sized bovid long bone flake. The scale bar in all samples equals 1 mm.

resting state (from near-vertical to oblique long-axis orientations (*Figure 9C*), suggesting a burial history involving multiple depositional events (*Manhein et al., 2006*).

## Skeletal part representation

With the exception of six avian bones and isolated non-diagnostic rodent elements, all identifiable, macro-skeletal specimens recovered to date from the Dinaledi Chamber are clearly hominin, and all the distinctive morphological aspects of *H. naledi* are near identical in excavated and surface-collected specimens and are considered part of the hypodigm of *H. naledi*. The collection of 1550 catalogued hominin specimens (*Berger et al., 2015*) includes ~1413 hominin fossil bone specimens,

**Table 1**. Element distribution patterns recording the Minimum Number of Individual and Indentifiable Elements (MNIE) for skeletal parts of the *H. naledi* assemblage from the Dinaledi Chamber

| Element | MNIE left | – | MNIE right |
|---|---|---|---|
| Mandible | | 7 | |
| Calvaria | | 6 | |
| C1 | | 2 | |
| C2 | | 2 | |
| Other cervical | | 3 | |
| Thoracic | | 13 | |
| Lumbar | | 3 | |
| First rib | 2 | | |
| Second rib | 1 | | 1 |
| Sternum | | 1 | |
| Clavicle | 3 | | 2 |
| Scapula | 2 | | 3 |
| Humerus | 3 | | 5 |
| Radius | 2 | | 4 |
| Ulna | 2 | | 3 |
| Scaphoid | 1 | | 3 |
| Lunate | 1 | | 2 |
| Capitate | 1 | | 2 |
| Trapezoid | 1 | | 2 |
| Trapezium | 1 | | 2 |
| Triquetral | | | 1 |
| MC1 | 4 | | 3 |
| MC2 | 3 | | 4 |
| MC3 | 3 | | 3 |
| MC4 | | | 1 |
| MC5 | 1 | | 1 |
| Proximal manual phalanges | | 35 | |
| Intermediate manual phalanges | | 27 | |
| Distal manual phalanges | | 14 | |
| Ilium | 4 | | 5 |
| Ischium | 4 | | 5 |
| Pubis | 3 | | 2 |
| Sacrum | | 1 | |
| Coccyx | | 1 | |
| Femur | 5 | | 9 |
| Patella | | | 4 |
| Tibia | 4 | | 5 |
| Fibula | 4 | | 4 |
| Talus | 6 | | 2 |
| Calcaneus | 2 | | 2 |
| Navicular | 3 | | 3 |
| Medial cuneiform | 3 | | 0 |
| Intermediate cuneiform | 3 | | 4 |

*Table 1. Continued on next page*

*Table 1. Continued*

| Element | MNIE left | – | MNIE right |
|---|---|---|---|
| Lateral cuneiform | 2 | | 1 |
| Cuboid | 1 | | 2 |
| MT1 | 3 | | 3 |
| MT2 | 1 | | 3 |
| MT3 | 2 | | 2 |
| MT4 | 1 | | 3 |
| MT5 | | | 3 |
| Proximal pedal phalanges | | 12 | |
| Intermediate pedal phalanges | | 14 | |
| Distal pedal phalanges | | 6 | |

including mandibles and maxillae bearing 53 teeth, and 137 isolated dental specimens. All dental crowns (n = 179), both from surface collection and in situ excavation, are hominin, and all are morphologically consistent with each other. Likewise, all morphologically informative bone specimens are clearly hominin. In all cases where multiple examples are present, they exhibit little variability, consistent with a single population sample. In general, the *H. naledi* assemblage contains many articulated elements, including articulated epiphyses of juvenile limb elements, and it contains a high number of infant and juvenile dental and postcranial remains (*Berger et al., 2015*). The remains represent a minimum of 15 hominin individuals, as indicated by the repetition and presence of deciduous and adult dental elements (*Berger et al., 2015*).

The avian bones (UW 101-40 a–f; *Aves* gen. et sp. indet.) represent lower limb elements (including femur, tibiotarsus and tarsometatarsus) consistent with coming from a single individual. The avian specimens were part of a group of bones that had been 'arranged' on rocks by an unknown caver prior to discovery by our caving team (*Figure 6B*), and are taphonomically distinct from the hominin assemblage. They lack surface modification and stain patterns, and are characterised by the adhesion of a thin film of calcite crystals over much of the surface of the bone, suggesting they were deposited more recently. A small number of isolated and non-diagnostic rodent incisors occur together with the hominin remains.

Using the bone elements recovered to date, we have analysed the composition of the *H. naledi* assemblage, based on the observed frequency of various elements as a function of the expected sequence of disarticulation of various skeletal regions, as well as intrinsic survival characteristics of elements (*Lyman, 1994*; *Pokines and Symes, 2013*). *Table 1* presents the Minimum Number of Individual and Identifiable Elements (MNIE) for the hominin skeletal material from the Dinaledi Chamber. This table is conservative and only lists elements that are clearly duplicated. Dental remains have not been included.

For long bone fragments, the MNIE is generally based on the repetition of the proximal shaft morphology (e.g., sub-trochanteric femur morphology), but, in some cases (e.g., the humerus) the MNIE is based on the distal articular morphology. In the current collection, this procedure underestimates the number of elements represented: for example, the total length of identifiable fibula shaft fragments is substantially greater than expected for the four left and four right fibulae that are duplicated in the assemblage based on proximal shaft morphology. Also, bone fragments, such as partial metacarpals, that are identifiable at regional scale, but that cannot be exactly identified at the scale of the element, have not been included in *Table 1*.

Whereas many long-bones are fragmentary, many of the hand, wrist, foot and ankle elements in the collection are complete, or nearly so. In several instances it is possible to identify immature and adult remains as representing duplicate elements even if they do not preserve the same anatomical area. We did not attempt an MNIE count with rib fragments beyond the first and second ribs; as such, there are 48 total rib fragments or refitted partial ribs aside from the two examples of first and second ribs.

What is most obvious from the general assemblage quantification is that all regions of the skeleton are represented within the cave assemblage, in spite of potential preservation and/or recovery bias. More importantly, in some of the material we observed, regions that conventionally disassociate both early and late in the decomposition process are represented as anatomically aligned and articulated groups, suggesting limited post-mortem spatial alteration and disaggregation within the chamber in both the proximate putrefactive (early) and distal decompositional (late) periods. Although much of the fossil material is disarticulated, the deposit contains articulated or near-articulated examples such as the maxilla and mandible of single individuals and the bones of the hands and feet, which normally disarticulate very early in the decomposition sequence (*Figure 9*). These elements are found in anatomical position and in spatial articulation with elements (e.g., vertebral components) that normally disarticulate later.

The observed patterns indicate a formational process that did not involve a high degree of winnowing. The sedimentary system in which the fossils were deposited was closed, or nearly closed, and skeletal disarticulation and movement of elements was largely restricted to the Dinaledi Chamber only (although some bone fragments may have washed down floor drains).

## Bone quality and preservation

The hominin assemblage is homogeneous in terms of surface preservation and condition (*Figures 10 and 11*), suggesting that the remains share a similar depositional history (i.e., the disarticulated vs articulated material does not vary significantly in terms of surface preservation). The structural state of the material is classified as good, and surface morphology is retained for many of the specimens (i.e., no plastic deformation has been observed), even though they were water logged and friable at recovery. The bones are generally, partially mineralised; there is no evidence of calcite crystal formation in or on bones, but some bones and teeth are dotted with black iron-manganese oxi-hydroxide deposits and coatings (e.g., *Figure 10*), and an orange-colour residue of iron oxide (e.g., *Figure 10B*). Colouration of the bone underlying surficial mineral deposits ranges from light grey to red-brown. The internal structure of bones is bright white in colour.

Whilst the overall structural integrity of the bone is good, few specimens preserve a pristine surface morphology. The overall surface quality of bone material corresponds to McKinley's Grades 3 and 4 (*McKinley, 2004*). As such, the majority of external surfaces of the cranial elements and long bones preserve the morphology of the bone, but display evidence of varying degrees of bone surface (cortical) removal (*Figure 11*; *Supplementary file 2*), striated patterns and pitting (*Figure 11*), and post-mortem fracturing resulting in multiple conjoins (*Figure 10*; *McKinley, 2004*)—expressions of these traits are discussed below.

## Weathering patterns

The skeletal assemblage of *H. naledi* displays little variation in surface structure and condition, indicating that the hominin material has been exposed to a limited range of environmental fluctuation during its depositional history. Analysis of the weathering patterns of the Dinaledi assemblage indicates that most of the bones show weathering stages 1 or 2 (after *Behrensmeyer, 1978*); while 18% have an etched appearance and 11% exhibit dissolution (*Supplementary file 2*). No element exceeds weathering stage 3 with the majority of elements displaying stage 1 weathering patterns. Most elements show no signs of cracking or flaking, although some deeper cracks may be present due to post-depositional effects. Stage 1 weathering is evidenced in most specimens of long bones, rib fragments, and mandibular fragments as fine longitudinal cracks (*Figure 10D*), without concomitant flaking and the formation of a fibrous texture.

None of the bone fragments studied preserve evidence of bleaching, cortical exfoliation, delamination or deep patination, indicating that the bones were not affected by solar radiation (*Lyman and Fox, 1989*, *1997*); that is, no bone fragment was exposed to weathering on surface outside the cave, which is consistent with sedimentation patterns observed in the chamber. We interpret the observed weathering patterns to result from fluctuations in moisture content in the Dinaledi Chamber, causing swelling and shrinkage of the osseous material, with larger longitudinal cracking being caused by sediment loading. As bone is repeatedly exposed to wet and dry conditions in seasonally/periodically wet environments, crack formation may be enhanced by the shrinking and swelling of the material (*Junod and Pokines, 2013*; *Pokines and Symes, 2013*). Overall, weathering patterns of the bone surfaces are consistent with the effects of sub-aerial and sub-surface processes in a periodically wet or water-saturated, dark depositional environment that experienced stable temperatures.

## Fractures

The *H. naledi* assemblage recovered from the Dinaledi Chamber is characterised by a high-degree of bone breakage and fragmentation (e.g., *Figure 10G*). Examination of the entire fossil assemblage for fracture patterns indicates that all of the fractures in the *H. naledi* assemblage resulted from dry bone (or mechanically incompetent) breakage, with many bones displaying recent or weathering related (19%) breakage patterns (*Villa and Mahieu, 1991*; *Symes et al., 2013*). Significantly, no examples of spiral (green) fractures have been documented (*Supplementary file 2*), which indicates that post-depositional processes are the primary agents of skeletal damage. Fracturing has led to disassociation and spatial dispersal of many of the skeletal elements (*Figures 6,7*). Refitting fragments of bone indicates that breakage occurred at different times in the depositional history of the assemblage. For some elements, neighbouring bone fragments display unique patterns of weathering and surface modification (e.g., *Figure 10H*); that is, breakage occurred relatively early in the depositional history and preceded periods of surface exposure. In other instances separate fragments that can be refitted to display matching staining patterns (e.g., *Figure 10F*) indicating that breakage occurred relatively late in the weathering history; that is, after the element had been lying on surface long enough for mineral stains to be deposited. These patterns are consistent with the geological observation that sediment in Units 2 and 3 with fossil remains were affected by reworking, with some bone material being re-deposited several times. We, therefore, interpret the observed dry-bone fracture patterns to be due to post-depositional sediment movement within the chamber as Units 2 and 3 are reworked, as well as unintentional damage by cavers or others entering the chamber; and we find no evidence for green fractures associated with trauma in the *H. naledi* fossil assemblage.

## Transportation

The matrix sediments surrounding the fossils do not present evidence for sedimentation processes that involved significant water action able to transport coarse-grained material including bone fragments. Likewise, the assemblage presents no evidence of fluvial modifications such as abrasion patterns caused by the movement of bone in water. No element shows evidence of physical smoothing or rounding, the development of polish or frosting, or indications of pitting, denting, and chipping caused by impact (*Supplementary file 2*; *Pokines and Symes, 2013*). Grooves, scratches, and gouges are common, but are the result of invertebrate action (see below; *Figure 11*), and the distribution of these defects across the bone surface is random. We therefore preclude transportation by water as a major taphonomic factor associated with the delivery of skeletal elements into, or within the cave system.

## Mineral staining

Staining and mineral deposition is common on most elements that preserve an external surface (*Supplementary file 2*). In the main, staining consists of irregular low-profile clusters or concretions of dark brown to black mineral deposit (a combination of manganese and iron oxy-hydroxides). This can occur as a surface deposit overlying the cortex directly (e.g., *Figure 10A*) and is seen as a surface sheen or cortical infiltrate, though more commonly it occurs as a deposit overlying existing areas of staining (yellow to ochre in colour (e.g., *Figure 10E*). This indicates a multi-staged history of mineralisation and chemical exchange between bone and the environment (*Pokines and Symes, 2013*), consistent with the episodic reworking of the bones.

A number of specimens preserve linear, and in some cases, parallel mineral precipitate stains or 'tide marks' that attest to the dynamic burial history of the fossils. These stains are visible as single or multiple linear deposits of manganese and iron oxy-hydroxide phases (*Figure 10F*), and are artefacts of the burial environment of osseous remains. They mark a contact boundary between the bone surface and surrounding sediment, and indicate the resting orientation of the bone during precipitation of the stains. They demonstrate that portions of individual bones were sub-aerially exposed for long periods of time during formation and erosion of Unit 3 (*Junod and Pokines, 2013*; *Pokines and Symes, 2013*). The outer and inner surface of the A1 cranium (UW 101–419) show at least two phases of cross-cutting mineral deposits, that formed when the cranium was complete (i.e., skeletonized and un-fragmented) and lying on its left side in contact with surrounding sediment that eroded and shifted over time. This clearly illustrates that at least part of the fossil assemblage in Unit 3 was reworked and redistributed as a result of erosion and re-deposition.

## Surface modifications and vertebrate–bone interactions

No evidence is noted of vertebrate modifications such as those caused by carnivores, rodents or other hominins. The specimens were assessed for evidence of edge polish from repeated gnawing, tooth pits or punctures perpendicular to the surface of the bone, tooth scores, striations and/or furrows (V or U-shaped in cross section), and traces of gastric corrosion (*Haynes, 1983*; *de Ruiter and Berger, 2000*; *Pickering et al., 2004*; *Pokines and Symes, 2013*; *Supplementary file 2*). There is no evidence of stone tool inflicted cuts, scrapes, impact or chop marks (*White, 2014*). Tooth scores and pits, crenulated edges and splintered shafts associated with carnivore damage (*Kuhn, 2011*) are absent. None of the specimens are burnt (*Stiner et al., 1995*) or shows signs of trampling other than limited incidental recent breakage by cavers that is readily evident (*Behrensmeyer et al., 1986*).

## Surface modifications and invertebrate–bone interactions

The collection bears clear traces of invertebrate activity, with most of the bones (n = 553) exhibiting microscopic removal of the bone surface, and evenly spaced, multiple parallel striations (35%) associated with smooth-based pits (34%), a pattern consistent with damage made to bone by modern gastropods. Sixty-two specimens (10%) record large individual, variably arrow-shaped and randomly oriented striations, consistent with those made by modern beetles (*Supplementary file 2*; *Figures 11,12*).

Many specimens in the assemblage show that the bone surface has been removed over a relatively large area, in various stages of elimination; from the outer lamellae to underlying cortical bone. Removal of the bone surface exposes underlying structures, enlarging features such as Haversian canals, and creating an etched appearance (e.g., *Figure 11B,E*). Fossil specimens retaining outer cortical lamellae show that a scraping mechanism was applied to the bone surface. This is indicated by the association of post-fossilisation incision marks with areas of bone surface removal (*Figure 11A*), and a bone fraction adhering to the fossil surface, its shape and distribution reflecting a scraping action.

Sets of shallow, evenly spaced, multiple parallel striations of equal width and length are a common feature on the fossils. When oriented parallel to the long axis of the bone, individual striations appear as relatively broad, variably arrow or lens-shaped. They follow the collagen fibres, creating an etched surface appearance. When oriented at an angle to the main axis of the bone, the morphology and arrangement of the parallel striations are more clearly visible, revealing multiple evenly spaced linear incisions, in places overlapping, and extending over a large area of the bone surface (*Figure 11B,E*). The multiple parallel striations locally manifest as short evenly spaced incision marks.

Shallow circular pits, ranging from about 0.1 to 3 mm in diameter are common. They are associated with bone surface removal (*Figure 11B,C*) and multiple parallel striations, in places in the base of pits. The margins of pits are generally smooth and rounded, and commonly scalloped, in a similar manner to the edge pattern observed on some outer cortical lamellae (*Figure 11C*). Some pits are fresh, exposed by recent removal of the sedimentary coating and fossilised bone surface. Straight edged scrape marks (*Figure 11D*) and clusters of large v-shave, overlapping striations (*Figure 11F*) are locally involved with bone surface removal.

The features observed on the Dinaledi collection were compared to surface modifications produced by 16 invertebrate taxa on bones in controlled experiments. Fresh, dry and fossil bones were placed with each type of invertebrate in separate tanks, and data on the resulting modifications recorded after four months. Our experiments show that land snails (*Achatina* sp.), garden snails (*Helix aspersa*), hide beetles (*Omorgus squalidus* and *Dermestes maculatus*) and their larvae produce a similar set of modifications to those observed on the Rising Star material, characterised by evenly spaced multiple parallel striations (*Figure 12A–C*), circular shallow pits with scalloped edges and smooth, cupped or striated bases (*Figure 12B,E*), scraping of the bone (*Figure 12D,F*), and an etched surface appearance (*Figure 12A,B*). This suggests that the observed surface modifications are most likely the result of gastropod and beetle damage.

At least two modification events have been observed on some of the fossils. Recent surface removal is evident on top of manganese and sedimentary fossil coatings, creating a white patina on bone (*Figure 11A*). Circular shallow pits and grooves, associated with multiple parallel striations and surface removal, are commonly in-filled with consolidated sediment (*Figure 11B,C*), indicating that the pits were created prior to the formation of the sedimentary coatings.

## Discussion

### Sedimentation processes in the Dinaledi Chamber and deposition of *H. naledi*

Whereas Unit 1, is a distinct older stratigraphic unit, Units 2 and 3 appear to have formed in a continual manner involving the interaction of three separate sedimentary processes: (a) sediment accumulation below access points into the cave chamber (i.e., near the current vertical entrance shaft); (b) erosion of the accumulating sedimentary pile, as sediment slumped down-slope, into the chamber towards floor drains; and (c) stabilisation by flowstone formation.

The accumulation of poorly consolidated Facies 2 sediment in Units 2 and 3, alternated with periods of flowstone deposition. Each of the Flowstones 1a-e interpreted to have formed as crust on top of a debris pile of Facies 2 sediment, starting with the formation of Flowstone 1a on top of a debris cone forming Unit 2. Formation of Flowstone 1a was followed by a period of erosion during which the poorly consolidated sediment pile, slumped inwards into the cave, probably as a result of the removal of sediment through floor drains deeper down in the cave chamber. This erosional process would have undercut the sediment pile of Unit 2, leaving behind partly indurated erosional remnants of Unit 2 covered by remnants of Flowstone 1a. As the sedimentary pile below this erosion remnant stabilized and flowstone formation continued, a new crust formed on top of the pile of Facies 2 sediment, to once more, weakly indurate the top-layer. This was followed by the next phase of erosion, slumping and undercutting of the sediment pile, leaving behind erosional remnants of Flowstone 1b again covering the remains of partly indurated Facies 2 mudstone breccia assigned to Unit 2. This progressive erosional process repeated itself at least three more times to form Flowstones 1c-e, as sediment of Unit 2 was gradually reworked and spread out across the floor of the Dinaledi Chamber, where it accumulated as Unit 3 (*Figure 4*). This process was concomitant with continued weathering, auto-brecciation and erosion of remnants of Unit 1, which also contributed to formation of Unit 3.

Deposition of Unit 3 was followed by a major phase of flowstone formation that formed cascades, curtains and flowstone crusts atop Unit 3 across the entire chamber (Flowstone 2). In the lower parts of the chamber and in many of the lower side passages, large areas of the cave floor (i.e., Unit 3) are still covered by Flowstone 2 (and to a lesser extent, more recent patches of Flowstone 3), including areas where the flowstone directly covers *H. naledi* bones. However, in the upper and central parts of the main chamber, Unit 3 has been locally eroded, undercutting and partly removing the cover of Flowstone 2; for example, near the fossil excavation pit, erosion remnants of Flowstone 2 along the chamber wall are positioned several cm's above the present chamber floor surface, indicating erosion of the uppermost part of Unit 3 since deposition of Flowstone 2 (*Figure 4F,G*). In addition, collapsed sediment containing hominin bones derived from hanging remnants of Unit 3 have locally washed down to be re-deposited atop Flowstone 2 lower down in the chamber, where the floor is level and no erosion has occurred. Thus, the erosional and re-depositional processes that led to the formation of Units 2 and 3, is on-going today, and the distribution of fossils on the cave floor is, in part at least, the result of erosional winnowing.

The remains of *H. naledi* are concentrated in Unit 3, but remains also occur in Unit 2. Four hominin bone fragments were observed in situ within Unit 2, and the shafts of a radius and ulna derived from a juvenile hominin were found in loose material slumped from hanging remnants of Unit 2. The large number of hominin bones collected from around the weathered base of Unit 2 also suggests that this unit contained a significant number of the fossils that contributed to the *H. naledi* assemblage.

However, many of the remains of *H. naledi* were probably not eroded out of Unit 2 to be re-deposited in Unit 3. Unlike Unit 2, which is limited to small erosion remnants near the cave entrance, Unit 3 is distributed across the Dinaledi Chamber and preserves well-articulated skeletal elements that have not been reworked from older units. This is illustrated by the presence of an articulated ankle, foot and hand in the excavation pit in Unit 3 (*Berger et al., 2015*this volume; *Figure 9*), which suggests that these remains were deposited in situ during the accumulation of Unit 3. It is highly unlikely that these remains weathered out from older sediments of Unit 2, because Unit 2 accumulated near the entrance to the chamber, that is, ~11 m away from the pit (*Figure 2C*), and there are no accumulations of Unit 2 above or near the excavation pit. In addition, if a hand or foot would have weathered out of the unconsolidated sediment of Unit 2, they would not have remained perfectly

articulated as the sediment slumped downslope into the chamber, past floor drains. Also, there have not been any flowstone fragments found within Unit 3 that would suggest that reworked skeletal material from unit 2 may have been transported *en bloc* deep into the chamber from the entrance area. Furthermore, the orientation of the chamber passages and presence of drain systems likely precludes such a possibility.

Thus, the fossils of *H. naledi* were probably deposited over an extended period of time during deposition and reworking of Units 2 and 3, and before deposition of Flowstone 2. Continual reworking of these units and the fossils they contain is consistent with taphonomic evidence (element distribution; breakage patterns, mineral staining). The distribution of Unit 2 below the current entry shaft into the Dinaledi Chamber, and the orientation of Flowstones 1a–e that cover Unit 2 and slope into the chamber, together strongly suggest that the current entry shaft has always been the main entry point for sediment into the chamber. This also indicates that the fossils of *H. naledi* entered via this route.

## Environmental constraints in the Dinaledi Chamber based on taphonomy

The Dinaledi collection displays taphonomic characteristics indicative of a depositional history that involved several stages of burial with surface modifications and breakage patterns consistent with repeated reworking of at least part of the assemblage within the confines of the Dinaledi Chamber, involving both biotic and abiotic agents (*Supplementary file 2*). The distribution of bone material and skeletal part representation indicative of limited winnowing (*Table 1*) indicate that the fossils of *H. naledi* must have found their way into the chamber via a difficult route that precluded any other large vertebrates from finding a way in. The distribution of the fossils within reworked material derived from Unit 2, as well-articulated remains in Unit 3 suggests that *H. naledi* fossils entered the chamber over an extended period of time; that is, not all remains were deposited at once. The presence of articulated, functional anatomical units that conventionally disassociate early in the decompositional sequence (*Figure 9B*) also suggests that bodies were fresh or in the early stages of decomposition when they entered the chamber and were encapsulated in the matrix (*Haglund, 1993*; *Lyman, 1994*). The lack of green fractures on any of the elements in the assemblage suggests that the bodies did not enter the chamber due to catastrophic accident such as falling into the chamber or due to flooding, or suffered trauma in any other way shortly before or after death.

Limited weathering (physical and chemical) indicative of sub-aerial, sub-surface processes in a periodically wet or water-saturated, dark environment (*Figure 10*) indicate that the bones were never exposed to the earth's surface and elements (the sun and rain) outside the cave (*Lyman and Fox, 1989*; *Backwell et al., 2012*; *Junod and Pokines, 2013*). The preservation of primary and secondary context patterns of skeletal elements, the fragmented nature of many fossil bones, overprinting tide marks of minerals stains on skull fragments (*Figure 10F*), invertebrate damage on bones by gastropods and beetles (*Figure 11*), and bone breakage patterns indicative of post-mortem (dry bone) or mechanically incompetent fractures (*Lyman and Fox, 1989*) probably due to re-working of some of the fossil-bearing sediments (*Figure 10*) all indicate that the fossils experienced significant periods of exposure and reworking after deposition (*Lyman, 1994*; *Manhein et al., 2006*). This multi-staged internment history is best understood in terms of the continual reworking of Units 2 and 3 due to the gradual erosion of the cave floor as it slumps toward floor drains in the chamber (the geological process explained above).

The collection is also notable for the absence of large vertebrate remains, and the taphonomic processes that are *not* evidenced through surface and structural modifications and skeletal part representation most importantly there is no evidence for carnivore or rodent modification; no evidence of cut marks or burns and no (geological or taphonomic) traces of fluvial transportation or long-distance movement or traces of green bone fractures (*Haglund, 1992*; *Lyman, 1994*; *Fernández-Jalvo and Andrews, 2003*; *Pickering et al., 2011a*).

## Depositional scenarios for the burial of *H. naledi*

Considering the geological and taphonomc context described above, the Dinaledi hominin site represents a depositional scenario that deviates from all other hominin localities in the region (*Brain, 1981*; *Partridge et al., 2003*; *Dirks et al., 2010*; *Pickering and Kramers, 2010*; *Pickering et al., 2011a*) in two important ways. First, the Dinaledi Chamber appears to have never had unimpeded

access, with fossil material deposited in muddy sediment in a geologically and geochemically isolated environment, and second the large vertebrate assemblage is comprised solely of hominin material (*Berger et al., 2015*). Each of these points will be discussed in more detail below.

Unlike other southern African cave sites where hidden shafts and sinkholes acted as death-traps to numerous species (*Brain, 1981*; *Partridge et al., 2003*; *Dirks et al., 2010*; *Pickering and Kramers, 2010*; *Pickering et al., 2011a*), there is no indication of a direct vertical passageway to surface into the Dinaledi Chamber. The chert layer forming the roof of the chamber is unbroken and coarse-grained clastic deposits transported by water action are absent, as are externally derived sediments. Instead, the Dinaledi Chamber represents an isolated sedimentary environment with fossils preserved in fine-grained, semi-consolidated mudstone breccia that is texturally and chemically distinct from fill in the upper chambers of the cave and that appears to have largely accumulated below the current entry shaft into the chamber.

Flowstone formation continues today (Flowstone 3), changing the morphology of cave passages. This makes it possible that a more direct access-way or easier passage may have existed when hominins entered. A different entrance into the chamber may also explain the presence of rodent bone concentrations in Facies 1b. However, sedimentation patterns indicate that the accumulation of Unit 2 with fossils occurred below the current entry point into the chamber, and alternate routes did not involve vertical access shafts that connected directly to surface in either the Dinaledi Chamber or nearby Dragon's Back Chamber. The lack of other contemporaneous fauna in the assemblage, and complete lack of surface modifications by vertebrates (carnivores, scavengers or rodents) further suggests that the Dinaledi Chamber remained undisturbed by other animals, which could not reach the chamber. Taken together, this means *H. naledi* would have had to traverse difficult terrain to reach the Dinaledi Chamber, with the chamber and the proximal entrance to the chamber positioned in the dark zone.

Mono-specific assemblages have been described from Tertiary and Mesozoic vertebrate fossil sites (*Kidwell et al., 1986*; *Rogers, 1990*; *Behrensmeyer, 1991*), linked to catastrophic events (*Turnbull and Martill, 1988*). Among deposits of non *H. sapiens* hominins, where evidence of catastrophic events is lacking, mono-specific assemblages have been associated typically with deliberate cultural deposition or burial (*Arsuaga et al., 1997*; *Carbonell and Mosquera, 2006*; *Gaudzinski, 2006*). Every previously known case of cultural deposition has been attributed to species of the genus *Homo* with cranial capacities near the modern human range. Unlike the Dinaledi assemblage, each of these hominin associated occurrences also contains at least some medium- to large-sized, non-hominin fauna (*Arsuaga et al., 1997*; *Carbonell and Mosquera, 2006*; *Gaudzinski, 2006*; *Behrensmeyer, 2008*).

There are a number of sites in Europe that represent accumulations of hominin fossils in caves that can be compared with the Dinaledi assemblage. The fossil hominins from Sima de los Huesos, Atapuerca, Spain, are presently believed to have accumulated within a single event (*Aranburu et al., 2015*). The assemblage also includes carnivore remains, including the remains of (an extensive number of) cave bears and smaller carnivores. The Sima de los Huesos bone assemblage is highly fractured, with most fractures consistent with post-depositional breakage, but a small portion (∼4%) consistent with peri-mortem breakage that might be explained as a result of fall-down a 13 m shaft into the cave (*Sala et al., 2015a*). Alternatively, there is some evidence that peri-mortem trauma may have resulted from lethal interpersonal violence (*Sala et al., 2015b*). Both hominin and carnivore bones in the Sima de los Huesos bear carnivore tooth marks at low frequency (less than 4%), which has been interpreted as the sporadic action of carnivores trapped after falling into the chamber (*Sala et al., 2014*). Cut marks have not been reported on the hominin remains (*Andrews and Jalvo, 1997*).

Another example is presented by Level TD6-2 of Gran Dolina, Spain, which represents an accumulation of hominin and faunal bones in a ∼30 cm thick layer across an excavated extent of 7 m². The remains of at least six hominin individuals are present in this assemblage, with cut marks and evidence of intentional de-fleshing similar to associated fauna, as well as tooth marks from carnivores (*Saladié et al., 2014*). The accumulation of hominin bone is interpreted as the result of cannibalism, with later carnivore scavenging at a smaller scale. A similar instance of cannibalism appears to be evidenced at El Sidron, Spain. Here, an almost exclusively hominin assemblage comprises a minimum of 13 individuals bearing cut marks, percussion pitting, and conchoidal scars typical of intentional processing of carcasses (*Rosas et al., 2012*).

The Krapina rock shelter in Croatia represents the earliest-excavated large hominin sample. Here remains of a minimum of 23 individual Hominins were recovered with extensive associated fauna from several stratigraphic layers, with most hominin material coming from a single layer, the 'Homo Zonus'. Breakage patterns of the Krapina hominin assemblage are mostly consistent with post-depositional fracturing, including excavation damage and prehistoric fracturing due to sedimentary loading (*Russell, 1987a*). Many of the hominin bones bear cut marks, which have been variably interpreted as evidence of cannibalism or mortuary practices (*Russell, 1987b*; *Villa, 1992*).

Pliocene and Early Pleistocene occurrences of single-species hominin bone deposits in Africa thus far contrast with these European cave sites. The AL 333 assemblage from Hadar, Ethiopia includes the fragmentary remains of some 17 hominin individuals and fauna, which appear to have been redeposited within a shallow streambed after possibly having been subject to predation (*Behrensmeyer, 2008*). Malapa, in the Cradle of Humankind, South Africa has been interpreted as a 'death trap', with several partial hominin skeletons and associated fauna (*Dirks et al., 2010*). The Malapa hominin and faunal assemblages lack evidence of carnivore activity, and include a disproportionate fraction of climbing species, suggesting that access to the cave was formerly limited and possibly hazardous for ungulates (*Val et al., 2015*), but a broad array of faunal species have been recovered from the site.

The Dinaledi assemblage shows no evidence of accumulation by either cannibalism or carnivore activity (*Supplementary file 2*), making it different from El Sidron, the Aurora Stratum of Gran Dolina, AL 333, and Krapina. The Dinaledi situation shows similarities with the Sima de los Huesos assemblage in several respects. However, the Dinaledi hominins recovered to date are entirely free of cut marks, tooth marks, or peri-mortem fractures while each of these is present at low frequencies in the Sima de los Huesos sample.

Few previously recognised scenarios operating in South African caves could have produced the selectivity for hominin remains as observed in the Dinaledi Chamber (*Brain, 1981*; *de Ruiter and Berger, 2000*; *Dirks et al., 2010*); a depositional situation common in no species other than modern humans. Below we will briefly discuss five alternative hypotheses for the accumulation of the hominin material in the Dinaledi Chamber, presented in order of what we consider to be reverse likelihood.

## Occupation

There are no signals of occupation debris or evidence of occupation within the Dinaledi Chamber, or anywhere else in the Rising Star cave. Based on our current assessment, occupation would have required accessing the chamber in the dark through an entrance similar to the current one, through an area inaccessible to other medium- to large-sized mammals. Thus, if hominins were traveling to the chamber, it is assumed that they would almost certainly have required artificial light. Given the physical challenges associated with access into the chamber and the lack of evidence for longer-term use, we consider occupation an unlikely explanation for the presence of the hominin remains.

## Water transport

Sedimentological evidence excludes the transport of coarse-grained material into the chamber by water action. The cave has been inundated in the past by rising water tables, but there is no evidence within the Dinaledi Chamber of depositional processes that involved high-energy transport mechanisms capable of transporting the large hominin bones, let alone do so in a uniquely selective manner. The high abundance and diversity of predominantly non-hominin fossil remains preserved throughout the adjacent Dragon's Back Chamber not only refute sediment transport by water flow between the two chambers, but also indicates that a considerable barrier between the two chambers was in place at time of deposition of *H. naledi*, and indeed throughout the history of the Dinaledi Chamber. The Dinaledi Chamber has been a closed depositional system for a long time, and did not allow the sudden ingress of water and sediment; only fine-grained muddy sediment accumulated, which accessed the chamber through narrow cracks that filtered out coarser material. Thus, the accumulation of the hominin remains in the chamber does not fit the pattern of a flood or fluvial event (*Behrensmeyer, 2008*; *Dirks et al., 2010*).

## Predator accumulation

In this hypothesis a carnivore either killed or scavenged the hominins, and brought them into the Dinaledi Chamber. In doing so, the predator would need to overcome the challenges of navigating the dark zone of the cave described earlier, in order to access a remote chamber, all while

transporting a large hominin carcass. Despite abundant fossil material available for taphonomic study, we have, thus far, found no trace of carnivore damage on the Dinaledi remains. Nor have we found any trace of carnivore remains or the remains of other likely prey animals. Thus, the predator would have had to select a single prey species—*H. naledi*—carrying into the chamber all age and size categories (*Berger et al., 2015*) without leaving a trace of its own presence. We consider this very unlikely.

## Mass fatality or death trap

As with other mono-specific assemblages in the fossil record (*Kidwell et al., 1986*; *Rogers, 1990*; *Behrensmeyer, 1991*, *2008*) the remains of *H. naledi* could have accumulated as a result of a catastrophic event during which a large group of animals was trapped in the cave. This could have happened either during a single event when a large number of hominin individuals were in the chamber, or in a death trap scenario over a period of time as individuals repeatedly entered the Dinaledi Chamber and died. Either scenario would have to explain why the animals chose to penetrate this deep into the cave, into the dark zone, moving away from all entrance points into the cave system. The sedimentological evidence presented suggests that accumulation of the fossils occurred over a period of time during deposition and reworking of Units 2, and 3, which refutes a single event hypothesis. Apart from this, and noting that the assemblage recovered to date represents only a small part of the total fossil content in the chamber, the sheer number of remains encountered in the Dinaledi Chamber, is hard to explain as the result of a single calamity. The individual entry hypothesis would require individuals or small groups to enter repeatedly and succumb to some form of, as yet unidentified, mortality event. The demographics of the assemblage—which includes individuals of practically every developmental age, from neonate to senile, is inconclusive in terms of providing definitive evidence towards either attritional or catastrophic mass fatality events.

The individuals represented within the Dinaledi chamber died at a range of ages from infant to old adult. The distribution of age-at-death within a sample of remains may potentially test hypotheses about the causes of mortality. For example, attritional mortality tends to over-represent old adults and very young children in comparison to their proportions within a living population, while the age-at-death distribution resulting from a catastrophic event tends to be representative of the age structure in a living population with more young adults and older juveniles than the attritional profile. The mortality profiles of the Sima de los Huesos and Krapina hominin samples have been argued as consistent with a catastrophic profile using statistical tests (*Bocquet-Appel and Arsuaga, 1999*). Most of the information to distinguish catastrophic from attritional age profiles in an age-at-death distribution is contained in the proportion of older juveniles and sub-adults. In both the Krapina and Sima de los Huesos assemblages, sample sizes are small and near the numerical limit to test statistically. In the Dinaledi assemblage, we presently can assess the age class (and, therefore, the approximate age-at-death distribution) for only 13 individuals, with 3 infants, 3 young juveniles, 1 old juvenile, 1 sub-adult, 4 young adults and 1 old adult (two additional individuals are represented by isolated teeth that cannot be attributed to an age class) (*Berger et al., 2015*). We found no significant result when comparing the currently available distribution to either catastrophic or attritional mortality profiles, and therefore a mass death scenario involving some sort of calamity or death trap cannot be completely excluded to explain the Dinaledi assemblage. The large number of immature individuals (8 out of 13) does allow us to reject hypotheses that would strongly over-represent adults, such as repeated cave exploration by socially isolated adult males. Further work in this regard will be required.

## Deliberate disposal

In the deliberate body disposal hypothesis, bodies of the individuals found in the cave would either have been carried into, or dropped through an entrance similar to, if not the same as, the one presently used to enter the Dinaledi Chamber. If individuals were dropped either whole or in part into the present entrance chute to the chamber, then physical entry would not be required. None of the bone elements studied shows evidence of green fracture (*Supplementary file 2*), indicating lack of trauma. Therefore, if bodies were dropped down the entrance, it is unlikely that they would have fallen rapidly, or landed with any force; perhaps because the entry is too irregular and narrow to allow a body to freefall and gain speed, or perhaps because a pile of soft muddy sediment had accumulated below the entry way, breaking the momentum of any falling object. Note that accessing the entry point to the chamber to drop bodies down the chute would have still entailed a complex climb in the

dark zone. In this scenario, the distribution of skeletal material, as well as the evidence for partial articulation, could be explained by the slow slumping of fleshed or semi-fleshed remains, downslope into the chamber. Alternatively, the hominins could have entered the chamber directly, carrying the bodies or dying there, which would explain, not only the absence of green fractures, but the presence of delicate, articulated remains in the excavation pit, deep in the chamber, well away from the entrance point, on the other side of floor drains.

Considering the geological and taphonomic context of the Dinaledi Chamber, the occupation, predator accumulation and water transport hypotheses cannot adequately explain the fossil assemblage. Both the mass mortality or death trap scenario (although possibly not involving a single event) and deliberate disposal hypothesis are considered plausible interpretations and require additional investigation. Based on current evidence, our preferred explanation for the accumulation of *H. naledi* fossils in the Dinaledi Chamber is deliberate body disposal, in which bodies of the individuals found in the cave would either have entered the chamber, or were dropped through an entrance similar to, if not the same as, the one presently used to enter the Dinaledi Chamber. Reconstructions of the cave environment indicate that reaching even the entrance of the Dinaledi Chamber would always have been difficult, particularly in the absence of artificial light.

Our interpretation of events raises questions about the meaning of deliberate and repeated body disposal to this ancient group of individuals. Recent evidence has extended the record of complex behaviour from archaic and modern humans (*Haglund, 1993*; *Arsuaga et al., 1997*; *Carbonell and Mosquera, 2006*) to earlier hominins (*McBreaty and Brooks, 2000*; *Douka and Spinapolice, 2012*; *Joordens et al., 2015*). Deliberate disposal of bodies in the Dinaledi Chamber implies that morphologically primitive hominins like *H. naledi* may have had their own distinctive patterns of behavioral complexity, even though the reason why *H. naledi* may have ventured deep into the cave system remains unresolved.

This leaves the important question of how old the *H. naledi* remains are. At this point we do not want to speculate on the age of the deposit considering the reworked nature of the sediments resulting in mixed stratigraphic signatures that impede faunal dating of the fossil rodent remains, and the limited amount of clean flowstone suitable for U-Pb dating (*Pickering et al., 2011a*, *2011b*). Further method development is underway to circumvent this problem.

## Materials and methods

### Discovery history of the Dinaledi Chamber and mapping of the cave system

The original mapping of the Rising Star cave system was conducted by a group of cavers from the Free cavers and CROSA caving societies in Johannesburg, who produced a base map that was used by our caving team when exploring the cave system. This map did not show the location of the Dinaledi Chamber.

The Dinaledi Chamber was discovered as a result of detailed speleological surveys of a series of chambers known to contain macro-fossils including the Dragon's Back Chamber. Once discovered, the Dinaledi Chamber and access routes into the chamber were mapped in more detail using traditional mapping techniques involving tape-measures, compass-clinometers, and laser-inclinometers. Prior to our work there is no evidence that excavations for fossils were ever undertaken in the Rising Star cave.

When the Dinaledi Chamber was first entered, it was clear that cavers had been in the chamber before, because they had re-arranged some of the bones (*Figure 6B*), and had left behind several survey pegs. It is unknown to what degree earlier caving expeditions may have disturbed the original context of the fossils, or damaged some of the bone material, by walking over them, although it was noted that upon first entry much of the rubbly floor of the Dinaledi Chamber had remained undisturbed (i.e., not walked on before). None of the earlier expeditions into the Dinaledi Chamber have left a record of the chamber itself on any survey maps, or have mentioned the fossils in the chamber. Further investigation has established only one confirmed entry into the chamber in the early 1990s (responsible for leaving the survey pegs). Given the complex nature of the entryway, it is unlikely that many, if any, expeditions would have entered the Dinaledi Chamber before or after that one confirmed entry. Since the discovery of the fossils, the entrances into the Dragon's Back Chamber have been locked off, and entry by recreational cavers is no longer physically possible.

## Mineralogical and geochemical analyses of floor sediments

Analytical work was carried out at the SPECTRUM central analytical facility of the University of Johannesburg. Grain-size analysis and petrography, were carried out at James Cook University. Bulk chemical analyses of 4 samples were carried out by X-ray fluorescence (XRF) using a Philips PANalytical MagiX Pro instrument and standard borate fusion (*Figure 5B*). The bulk mineralogical composition of four samples was determined by X-ray diffraction (XRD) using a Philips PANalytical X'Pert instrument. The High Score Plus software was used to refine the XRD diffractograms to identify the various mineral phases within the sediment samples.

The cave sediment samples were mounted in 30 mm epoxy blocks and polished for textural analysis using scanning electron microscopy (SEM; *Figures 5, 8*) and electron microprobe analysis. SEM studies were carried out using a Tescan Vega3 scanning electron microscope equipped with an Oxford Instruments X-max 50 mm$^2$ EDS detector. Quantitative spot chemical analyses were carried out using a CAMECA SX100 electron microprobe with 4 wavelength dispersive spectrometers and an EDS detector. The instrument was operated at 15 kV with a 5 µm beam width. As many phases within the fragments constituting the samples are not resolvable even at the µm scale, most of the analytical results represent mixtures, and information on their constituent minerals can be obtained from element correlations. The data are given in *Supplementary file 1*. Relevant two-element plots are shown in *Figure 8*. For the quartz-dominated sample DB-1 from the Dragon's Back Chamber, the non-quartz components were investigated.

## Excavation approach

Following discovery of the fossil deposits, permission for excavation and recovery of fossils was issued by the South African Heritage Resource Agency, Department of Arts and Culture (PermitID: 952). Excavations are coordinated through the Evolutionary Studies Institute and the National Centre for Excellence in PalaeoSciences, at the University of the Witwatersrand, where all specimens are curated and stored in the fossil vault, and can be studied and accessed by researchers.

Due to the difficult operating conditions, and limited physical access to the excavation chamber by the field crew, a unique system of recording and recovery was developed to assist excavation in the Dinaledi Chamber. The excavation and recording strategy focused on maximizing the retrieval of stratigraphic and spatial information during the recovery process by the excavators, and at the same time allowing reflexive supervision from senior investigators in a physically remote location. The technical strategy included the use of high-resolution 3D non-contact scanning, live digital video streaming of the excavation process to an above-ground supervisory team, as well as more conventional archaeological recording methods to facilitate post-excavation analysis.

Although the recovery process utilised conventional archaeological methodologies, an explicitly forensic archaeological approach (*Hunter and Cox, 2005*) was applied to the resolution of the Dinaledi Chamber. We adopted a multidisciplinary framework, bringing a wide range of expertise in buried environments, to ensure that the most complete range of evidence was collected. The excavation strategy proceeded on a single-context recording and recovery basis (*MOLAS, 1994*). The limits of the excavation were defined within the site, and sketch plans produced where appropriate. Excavation was undertaken with non-metallic tools in order to limit the possibility of recovery damage to highly fragile and friable skeletal material, and the exposure and recovery of any sediment surface was limited to the production of one or two hand-brush trays of spoil only. All excavated matrix sediment was double-bagged and recovered for analysis. In situ metric measurements were taken where specimens were highly fragmented or fragile. Taphonomic traces were noted where seen. All bone fragments were lifted from the surrounding sediment, double bagged, wrapped in bubble plastic, taped securely, boxed and lifted to the surface for cleaning, photographic recording, and consolidation or preservation of the remains where necessary. Recording pathways adopted are presented in *Table 2*.

A photographic record of the excavation was taken, augmented with high-resolution, 3D surface scanning to document the location, orientation (axial and surface) of every bone. We used an Artec Eva (http://www.artec3d.com/hardware/artec-eva/), 3D white light scanner with the capacity to capture surface colour and texture (surface resolution: 0.5 mm; 3D point accuracy: 0.1 mm) in order to record and analyze the spatial position and orientation of bones within Unit 3 matrix material. The post-scan process was managed in Artec Studio 9. Each scan sweep comprised a number of separate

**Table 2.** Recording activity pathway for Dinaledi Chamber excavations

| Material or action | | Assign and record | Forms |
|---|---|---|---|
| In cave: excavator and recorder | | | |
| Sediment context | | Context number and attributes | Context form |
| | | Photography | Photo register |
| | | Sketch plan and/or section | Section log |
| | | Scan | Scan register |
| | | Sample | Sample register |
| Bone | | Element number | Skeletal form |
| | | Spatial properties | Area sketch |
| | | Physical properties | Taphonomy |
| | | | Peri-M trauma |
| | | | Post-M trauma |
| | | | Metric form |
| | | Recovery | Recovery log |
| Excavation scan | | Record all observed contexts and elements | Scan form |
| Above ground: recorder | | | |
| Context or element | | Record of assigned number | Context log |
| Excavation scan | | Record all observed contexts and elements | Scan log |
| Bone element | | Preliminary ID, spatial attributes and location | Element log |
| Recovered material | | What is lifted and boxed | Recovery log |

images compiled into a single layer. Each scan was then registered to acquire 3D triangulated points. Once registration was complete, the separate layers were manually aligned, by using a minimum of three reference points. The reference points are fixed survey markers within the Dinaledi Chamber, which were captured in each scan to provide optimal registration. Once alignment was complete, a global registration process allowed for the scan data to be merged accurately. This produced a 3D mesh representation of the scanned area. The 3D scan was then overlain with a photographic texture map, which was captured by the Artec Eva at the time of scanning.

## Methodology of taphonomic analyses

The entire *H. naledi* assemblage has been analysed at macroscopic scale, and a reduced sample of specimens from 11 individuals has been studied by microscope. Specimens were viewed at magnifications between 7 and 50 times using an Olympus SZX16 Zoom Stereo Microscope fitted with a DFC420 digital camera and equipped with Stream software. Modifications observed on the Rising Star material were compared with those recorded on bones in a reference collection held at the University of the Witwatersrand, which comprises bones and teeth modified by 58 known agents, including humans and a wide range of other vertebrates; 17 invertebrate taxa; and geological processes, including different forms of sedimentary abrasion, from trampling by large animals to the overall rounding and polishing caused by moving water. Characteristics and traces evaluated in the taphonomic analysis are listed in *Table 3* (after *Pokines and Symes, 2013*).

In undertaking the analyses we have adopted a taphonomic approach derived from forensic practice in relation to the death process and burial environment with a degree of resolution usually reserved for medico-legal casework. The co-option of forensic taphonomy into palaeo-taphonomy provides a framework with which to investigate the decompositional and formational histories of ancient deposits by applying analyses of short-period events into the geological past (*Symes et al., 2013*).

**Table 3.** Taphonomic recording criteria (after *Pokines and Symes, 2013*)

| Signature | Characters or taphonomic traces for recording |
|---|---|
| Preservational | General state of remains (excellent, good, fair, or poor) |
| | Cortical erosion/exposure of cancellous bone |
| | Cortical exfoliation (bone loss in thin, spalling layers) |
| | Postmortem breakage |
| | Perimortem breakage/fragmentation or trauma |
| | Rounding (erosion/tumbling in an abrasive environment) |
| | Decalcified |
| | Postmortem cracking of desiccated tooth enamel |
| | Incidental surface striations/scratches |
| Soil surface exposure | Surface cracking/longitudinal splitting from drying of waterlogged bone |
| | Weathering (bleaching and cracking; sensu Behrensmeyer) |
| Mineral deposition | Copper (green), iron (red), calcium (white), manganese (black), or other mineral oxide staining |
| | Vivianite formation |
| | Concretion |
| | Water staining (presence of a water line from mineral deposits, colour differential line) |
| Mechanical | Excavation damage |
| | Micro-abrasion |
| Soil/burial substrate | General soil staining |
| | Warping/flattening of elements (especially the cranial vault) |
| | Crushing/compaction from overburden |
| | Adhering/infiltrating sediments |
| Faunal | Adhering fauna |
| | Carnivore puncture and gnawing |
| | Gastric corrosion, winnowing, or windowing of bone |
| | Rodent gnawing |
| | Invertebrate surface modification and damage |

## Description of weathering patterns

The remains of *H. naledi* were analysed following the six stage classification system of *Behrensmeyer (1978)* based on faunal remains that weathered on the landscape in Kenya. Although this classification system has been developed for surface deposits, and is therefore less suitable for South African cave deposits, we have chosen to apply this recording system, because it allows us to address issues of possible bone transport into the Dinaledi Chamber from surface. *Behrensmeyer (1978)* described six weathering stages (Stages 0–5) based on the progressive pattern of linear cracking and flaking of the cortical surface, followed by formation of a rough fibrous texture, and eventual collapse of bone integrity and structure (i.e., greasy bone with tissue is designated as Stage 0, and splintered fragile bone falling apart in situ is designated as Stage 5). The progressive weathering stages are broadly indicative of the period of surface exposure in sub-aerial conditions, with bone exposed to the elements, including sunlight.

## Acknowledgements

We would like to thank the many funding agencies that supported various aspects of this work. In particular we would like to thank the National Geographic Society, the National Research

Foundation and the Lyda Hill Foundation for significant funding of the discovery, recovery and analysis of this material. Further support was provided by ARC (DP140104282; PHGMD; ER), and by the TAMU College of Liberal Arts Seed Grant Fund (DJdeR) and the Palaeontological Scientific Trust (PAST). We would also like to thank the University of the Witwatersrand and the Evolutionary Studies Institute as well as the South African National Centre of Excellence in PalaeoSciences for curating the material and hosting the authors while studying the material. We would like to thank the South African Heritage Resource Agency for the necessary permits to work on the Rising Star site and the Jacobs family for granting access. The assistance of members of the Speleological Exploration Club, in various safety aspects within the cave during excavations is gratefully acknowledged. Contributions to mapping the original cave system by Harris et al. (1985) from the Free cavers and CROSA caving societies, are acknowledged. We would also like to thank Wilma Lawrence, Bonita De Klerk, Natasha Barbolini, Merrill van der Walt, Wayne Crichton and Justin Mukanku for their assistance during all phases of the project. The assistance of Alexander Parkinson and Jennifer Randolph-Quinney is gratefully acknowledged in the production of the supplementary online materials.

## Additional information

### Funding

| Funder | Grant reference | Author |
|---|---|---|
| National Geographic Society | | Lee R Berger |
| National Research Foundation | | Lee R Berger |
| Lyda Hill Foundation | | Lee R Berger |
| Australian Research Council (ARC) | DP140104282 | Paul HGM Dirks, Eric M Roberts |

The funders had no role in study design, data collection and interpretation, or the decision to submit the work for publication.

### Author contributions

PHGMD, Responsible for geological mapping, wrote the original draft of the manuscript, and coordinated all subsequent edits, and assisted with overall interpretations; LRB, Conceived the overall project, assisted with taphonomic interpretations and wrote interpretive parts of the manuscript; EMR, Mapped the geology and lay-out of the Dinaledi Chamber, provided sedimentological input and assisted with the interpretations and geological writing; JDK, Provided the geochemical and sedimentological input and assisted with the interpretations; JH, Analysed the spatial context and skeletal distribution patterns of elements; Provided input on interpretations of behavior and taphonomy; Helped draft sections of the discussion; PSR-Q, Contributed to the excavation and body recovery protocols and drafted the taphonomic sections; Conducted analyses of weathering, skeletal damage patterns and other taphonomic processes in deposition and formation of the assemblage; ME, Conducted the excavations and fossil recovery, supervised by JH and LRBe; Assisted with detailed sampling and cave mapping; CMM, Conducted analyses of weathering, skeletal damage patterns and other taphonomic processes in deposition and formation of the assemblage; SEC, DJR, PS, Assisted with the interpretations related to behavior; LRB, Provided input on surface modification and with JGH conducted experiments on invertebrate–bone interactions; Conducted analyses of weathering, skeletal damage patterns and other taphonomic processes in deposition and formation of the assemblage; GAB, Provided the geochemical and sedimentological input and assisted with the interpretations; PB, Helped map the cave system; KLH, Conducted the excavations and fossil recovery supervised by JH and LRBe, Acquisition of data; EMF, AG, HM, BP, Conducted the excavations and fossil recovery supervised by JH and LRBe; JGH, Conducted experiments on invertebrate–bone interactions; RH, ST, RH and ST were the cavers who discovered the fossils, and together with PB mapped the cave system; AK, Helped with the analyses of the spatial context and skeletal distribution patterns; TVM, Collected geochemistry data on sediments

## Additional files

### Supplementary files

• Supplementary file 1. Electron microprobe analyses of spots in fragments of samples UW101-SO-31, UW101-SO-34, UW101-SO-39 and DB-1. Note that in each of the tables totals below 100% reflect volatile content or porosity of sample, or both.

• Supplementary file 2. Summary table listing surface modifications on all morphologically informative specimens. A total of 559 bone and dental specimens were examined for surface modifications. This sample includes all of the larger specimens and most of the complete elements in the collection, from both surface and excavation contexts. At low magnification (7×) most of the bones show weathering stage 1 or 2 (*Behrensmeyer, 1978*), while 18% have an etched appearance and 11% exhibit dissolution. No bone in this sample has edges consistent with a spiral (fresh) fracture. Instead, they show weathered (19%) or recent breakage patterns (*Villa and Mahieu, 1991*). None of the specimens are burnt (*Stiner et al., 1995*) or shows signs of trampling (*Behrensmeyer et al., 1986*). There is no evidence of stone tool inflicted cut, scrape, impact or chop marks (*White, 2014*). Tooth scores and pits, crenulated edges and splintered shafts associated with carnivore damage (*Kuhn, 2011*) are absent. The collection bears clear traces of invertebrate activity, with most of the bones (n = 553) exhibiting microscopic removal of the bone surface, and evenly spaced, multiple parallel striations (35%) associated with smooth-based pits (34%), a pattern consistent with damage made to bone by modern gastropods (*Figure 12*). Sixty-two specimens (10%) record large individual, variably arrow-shaped and randomly oriented striations, consistent with those made by modern beetles (*Figure 12*).

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
