## [Decision Letter]

Thank you for submitting your work entitled “Geological and taphonomic evidence for deliberate body disposal by the primitive hominin species *Homo naledi* from the Dinaledi Chamber, South Africa” for peer review at *eLife*. Your submission has been evaluated by a Senior Editor (Ian Baldwin), two guest Reviewing Editors (Nicholas Conard and Johannes Krause), and two peer reviewers. One of the two peer reviewers, Palmira Saladié, has agreed to share her identity.

The reviewers have discussed the reviews with one another, and Nicholas Conard has drafted this decision to help you prepare a revised submission.

The Dinaledi Chamber site is potentially very interesting because of the amount of hominin fossils that have been recovered there. This paper describes the geological and taphonomic study of a cave site where more than 1500 hominin fossils have been recovered in two excavation campaigns. The fossils were found on the surface of the cave floor and in one excavation pit of 0.8x0.8m and 50 cm of maximum depth.

I share the reviewers' opinion on the great importance of the paper you have submitted to *eLife*. One said: “I believe that the description of the site is relevant for publication by itself since it is a new and very important site.” The other commented:

“The methods used in the investigation were correct, and their arguments are based on good evidence. The paper is clear and the set of fossils from the Dinaledi Chamber is obviously an exceptional assemblage in the world. It will certainly increase our understanding of human evolution, but also the knowledge about the evolution of human behavior many times so difficult to infer. The authors have inferred through taphonomic and geological studies that is an intentional accumulation of remains of hominins, located at a very early age. This hypothesis will not be free of possible controversy; however, empirical data that are available appear to suggest that one cannot point to any other optional hypothesis, in my view.”

I also share their opinion that we should view this paper as an initial report. The issues of hominin intentionality are controversial, and these along with other taphonomic questions will need a more detailed and critical assessment in the future.

For example, one reviewer said: “I believe that the conclusions provided in the manuscript submitted by the authors go too far in interpreting the data provided […]. Based on [the] limited evidence about the stratigraphy and its relationship to the fossils, it is premature to make detailed interpretations about the site formation processes. In fact, the stratigraphic section of the site presented in the article consists of an unscaled schematic diagram. At the same time, the main unit – where the fossils have been recovered – clearly consists of reworked sediments and no direct (geo-chronological) or indirect (bio-chronological) age has been provided for the deposits. Finally, the taphonomic assertions in the main text of the article [would] need to be supported by corresponding quantitative data (e.g. number and frequencies of each fracture attribute, comparison with other modern or archaeological assemblages, statistical analysis, etc).”

In this context, I recommend that you follow the constructive criticism of the reviewers and present the multiple possible interpretations as competing hypotheses that need to undergo more testing in the future. I have no problem with you stating your preference and your reasons for preferring this interpretation, but do focus on this as one of multiple explanations and give fair treatment to multiple points of view. I think this will defuse some of the controversy about the site while still allowing you to state your preferred interpretation.

Revisions required:

1) We strongly recommend that you revise the current manuscript into a descriptive article about the architecture of the cave, the preliminary geological analysis (stratigraphy, mineralogy…), the description of the fossiliferous deposit, and so on, leaving the interpretations about the origin of the fossil accumulation for forthcoming and more detailed papers. The authors have provided much relevant data regarding the description of the karst system, which is important to their interpretation.

On the other hand, if the authors wish to maintain the current title and interpretations of the paper, they must provide all the necessary data to support the taphonomic conclusions: e.g., fracture patterns of cranial and postcranial remains, bone surface modifications (at least frequencies), statistical comparisons of taphonomic features with modern and/or archaeological assemblages, elaboration of detailed stratigraphic cross-sections, etc. If the authors provide detailed data about the fracture patterns, the results should be compared with other assemblages, see for example Villa, P., Mahieu, E., 1991. J Hum. Evol. 21, 27–48. Hominin bone accumulations in karst systems are often very complex to study and interpret, and the Dinaledi Chamber is particularly interesting in this regard.

2) There are several examples of hominin fossils being found in cave systems (e.g. Atapuerca-Sima de los Huesos, Krapina, Peștera cu Oase). Some comparison with these other cases is warranted (or at the very least mentioned). How is the accumulation in the Dinaledi Chamber similar or different taphonomically from the accumulations at these other sites? The authors might choose a different set of sites, but some kind of comparison with the geological context and/or taphonomic aspects of other sites in the fossil record should be mentioned or discussed.

Other points to consider:

1) It would be interesting to know in greater detail the age of the individuals recorded. The authors indicate that there are individuals of all ages, from neonates to senile. However, this age profile may correspond with a case of catastrophic death. A better exposure of the age profile may help reject a single event of death.

2) Are many remains absent in the assemblage? I understand that this is a fragmentary and not fully excavated set, but it would be interesting to know the relationship between bone mineral density present to better appreciate the conservation of the remains (and the absence of the activity of different taphonomic processes and agents as carnivores or the weathering).

3) A better description of the bone breakage and their characteristics is desirable. In fact, if the bodies were thrown into the chamber, some bones could eventually present some perimortem fracturing.

---

## [Author Response]

In revising the paper, we have made the following changes:

We have changed the title, most notably by removal referral to purposeful burial.

We have toned down our interpretation in the Abstract and have allowed for alternative interpretations, although we still state our preferred interpretation.

We have extended the Introduction and combined a general introduction to cave deposits in the CoH in South Africa with an extended introduction to the Rising Star cave system.

The Discussion is presented in three parts:

a) A geological processes part, explaining the geological processes that caused deposition and reworking of the units containing the bones.

b) A taphonomy part, which looks at constraining environmental factors in the Dinaledi Chamber.

c) Depositional scenarios: we have added a long comparative section with caves in Europe, we have expanded the scenario section – especially the sections involving mass death (by including more info on the population structure) and deliberate disposal (by better explaining the absence of green fractures in this scenario) – and we have toned down our arguments for our preferred option, allowing more room for alternative interpretations. We have also removed speculation on possible dates for the fossils, and we have largely removed reference to the use of fire (although we still indirectly refer to it). All in all, we have taken a more cautious approach to interpreting the data and we have allowed more leeway for alternative interpretations. We have also indicated more clearly that additional work is required before definitive hypotheses can be put forward.

*In this context, I recommend that you follow the constructive criticism of the reviewers and present the multiple possible interpretations as competing hypotheses that need to undergo more testing in the future. I have no problem with you stating your preference and your reasons for preferring this interpretation, but do focus on this as one of multiple explanations and give fair treatment to multiple points of view. I think this will defuse some of the controversy about the site while still allowing you to state your preferred interpretation*.

We accept this criticism and in response we have done several things to make the text less controversial, and allow more room for alternative interpretations:

a) We have changed the title and removed reference to purposeful burial (focusing instead on geological and taphonomic context).

b) In the Abstract, we have softened the language around our preferred interpretation by stating: “The unique accumulation of *H. naledi* requires further study to fully understand, however preliminary evidence is consistent with deliberate body disposal in a single location”. This leaves the door open for alternative interpretations, which we have allowed for in the Discussion.

c) In the Discussion section, we have expanded the discussion on various alternative scenarios, especially those involving mass death events as a result of some sort of calamity or death traps vs purposeful entry and body disposal. In the Discussion we have included comparisons with European cave sites (allowing the reader to at least appreciate the possibility of the involvement of cannibalism or some other form of traumatic death trap scenario – for which we have found no evidence, thus far, in the Dinaledi assemblage), as well as several other sites with high concentrations of hominin bones (e.g. AL333, Malapa) to further illustrate the anomalous nature of the Dinaledi assemblage.

d) We have removed any speculation on dates for the fossils.

e) We still state our preference, but we have more clearly allowed for alternatives (especially the mass death scenario); we have stated that more work is needed and we have significantly toned down the more controversial statements.

*Revisions required*:

*1) We strongly recommend that you revise the current manuscript into a descriptive article about the architecture of the cave, the preliminary geological analysis (stratigraphy, mineralogy…), the description of the fossiliferous deposit, and so on, leaving the interpretations about the origin of the fossil accumulation for forthcoming and more detailed papers. The authors have provided much relevant data regarding the description of the karst system, which is important to their interpretation*.

Essentially this is what we have done; we would like to retain some of the interpretation, but we have significantly toned down the more controversial parts of our interpretations as explained above, and we have more clearly allowed for alternatives.

*On the other hand, if the authors wish to maintain the current title and interpretations of the paper, they must provide all the necessary data to support the taphonomic conclusions: e.g., fracture patterns of cranial and postcranial remains, bone surface modifications (at least frequencies), statistical comparisons of taphonomic features with modern and/or archaeological assemblages, elaboration of detailed stratigraphic cross-sections, etc. If the authors provide detailed data about the fracture patterns, the results should be compared with other assemblages, see for example Villa, P., Mahieu, E., 1991. J Hum. Evol. 21, 27–48. Hominin bone accumulations in karst systems are often very complex to study and interpret, and the Dinaledi Chamber is particularly interesting in this regard*.

As indicated above, we have not maintained the title and we have toned down the interpretations considerably. We have also included a table ([Supplementary-material SD2-data]) summarizing a full analysis of surface modifications including frequencies on all morphologically informative specimens (cranial and post-cranial). These results have been described and discussed, and show that all fractures are dry-bone fractures. No green fractures have been observed on any of the bone specimens recovered to date.

We recognize that more work will need to be done in future before full conclusions can be drawn, and we say so in the text. Therefore, we have significantly toned down the interpretations in the Discussion section leaving more space for alternative scenarios.

*2) There are several examples of hominin fossils being found in cave systems (e.g. Atapuerca-Sima de los Huesos, Krapina, Peștera cu Oase). Some comparison with these other cases is warranted (or at the very least mentioned). How is the accumulation in the Dinaledi Chamber similar or different taphonomically from the accumulations at these other sites? The authors might choose a different set of sites, but some kind of comparison with the geological context and/or taphonomic aspects of other sites in the fossil record should be mentioned or discussed*.

We have included text in the subsection “Depositional scenarios for the burial of *H. naledi*” which provides a comparative discussion between European cave sites (Sima de los Huesos, Gran Dolina, Krapina) and Dinaledi, and have also made a brief mention of AL333 and Malapa.

Other points to consider:

*1) It would be interesting to know in greater detail the age of the individuals recorded. The authors indicate that there are individuals of all ages, from neonates to senile. However, this age profile may correspond with a case of catastrophic death. A better exposure of the age profile may help reject a single event of death*.

We have included text in the subsection “Mass Fatality or Death Trap” to discuss the age profile of individuals recovered from the Dinaledi assemblage in more detail. We do so within the section where we discuss the various scenarios. The age-at-death profile is not conclusive when it comes to demonstrating or rejecting a single event of death, and we make this clear in the text.

*2) Are many remains absent in the assemblage? I understand that this is a fragmentary and not fully excavated set, but it would be interesting to know the relationship between bone mineral density present to better appreciate the conservation of the remains (and the absence of the activity of different taphonomic processes and agents as carnivores or the weathering)*.

This has been discussed in the taphonomy section under “Skeletal part representation” (where the data on MNIE is presented; this data is also summarized in Table 6). Not many remains are absent from the assemblage, there is limited winnowing, there is no evidence of carnivore activity, and weathering is limited.

*3) A better description of the bone breakage and their characteristics is desirable. In fact, if the bodies were thrown into the chamber, some bones could eventually present some perimortem fracturing*.

We have now included a table summarizing a full analysis of surface modifications including fracture frequencies on all morphologically informative specimens (cranial and post-cranial). These results have been described and discussed and show that all fractures are dry-bone fractures. Amongst the almost 600 samples studied, no green fractures have been observed on any of the bone specimens recovered to date.